# Asymptotic Optimality of the High-Dimensional Gaussian Mechanism and Improved Low-Dimensional Mechanisms for Differential Privacy

**Yu Wei** [1 2]   **Alexander Bienstock** [3]   **Antigoni Polychroniadou** [3]

## Abstract

The additive noise mechanism is a foundational tool for differential privacy (DP) of $T$-dimensional real-valued vector queries. The Gaussian mechanism, utilizing Gaussian noise, is the mostly widely used such mechanism, due to its simplicity and strong privacy guarantees. In this work, we provide justification for this choice, showing that as the dimension $T \to \infty$, no additive-noise mechanism can asymptotically improve on the Gaussian mechanism's privacy–utility tradeoff for the strong privacy settings typically used. We also develop a new family of *Spherical Generalized Gamma* DP mechanisms, which contains both the Gaussian mechanism and the recently studied $\ell_2$ mechanism (Joseph *et al.*, ICML 2025). We identify members of this family that outperform both the Gaussian and $\ell_2$ mechanisms in certain low-dimensional settings, and show tight composition of all mechanisms in this family, answering an open question of Joseph *et al.* regarding the $\ell_2$ mechanism.

## 1. Introduction

Differential Privacy (DP) (Dwork et al., 2006b) is a powerful tool that can be utilized to extract aggregate insights from data, while retaining individuals' privacy. Differential privacy has myriad privacy-preserving applications, including training deep learning models (Abadi et al., 2016), generating synthetic data (Kurakin et al., 2024; Zhang et al., 2021; Torkzadehmahani et al., 2019; Yoon et al., 2019; Mckenna et al., 2019), publishing aggregate statistics (Dwork et al., 2006b), and more.

Many applications of DP, including the above, involve protecting real-valued $T$-dimensional vector queries $Q(D) \in \mathbb{R}^T$ over private datasets $D$. In this setting, we want to obfuscate any addition or removal of a single data record from $D$, which may perturb $Q$ by any $\mu \in \mathbb{R}^T$. Typically, the size $||\mu||$ of $\mu$ is bounded by some scalar $s$, the *sensitivity* of the query. The standard way to achieve DP in this setting is by using the seminal additive noise mechanism (Dwork et al., 2006b). This mechanism simply adds $T$-dimensional noise $X$ sampled from some distribution $\mathcal{D}$ to the query: $\widehat{Q} = Q + X$. The typical distribution used is the multivariate Gaussian distribution (Dwork et al., 2006a; Dwork & Roth, 2014), i.e., $X \sim \mathcal{N}(0, \sigma^2 I_T)$, where $\sigma \propto s$. This *Gaussian mechanism* has several appealing properties, including (i) privacy with respect to $\ell_2$-sensitivity, as opposed to e.g., $\ell_1$-sensitivity, which is important to retain utility of high-dimensional queries (ii) spherical symmetry, which means that privacy does not depend on the direction of the query sensitivity, (iii) tight, closed-form privacy analysis (Balle & Wang, 2018b), and (iv) tight composition guarantees (Dong et al., 2022; Bun et al., 2018).

Even though the Gaussian mechanism has these desirable properties and has found wide use in practice (Bittau et al., 2017; Apple, 2016; Google, 2023), one may wonder whether it is the *best* mechanism for privately answering real-valued vector queries. Indeed, prior work has explored alternative mechanisms for such queries: It was recently shown that the so-called $\ell_2$-mechanism (Joseph et al., 2025) (essentially, a high-dimensional version of the Laplace mechanism (Dwork et al., 2006b) for $\ell_2$ sensitivity) outperforms the Gaussian mechanism in certain low-dimensional settings, while appearing to obtain similar performance in higher dimensions.

However, these two questions remain to be answered:

(Q1) Is the Gaussian mechanism *optimal* in high dimensions?

(Q2) Are there mechanisms that outperform *both* the Gaussian and $\ell_2$ mechanisms in low dimensions?

**Our Contributions.** In this paper, we answer both questions affirmatively, through the lens of $(\varepsilon, \delta)$-DP (Dwork

[1]Georgia Institute of Technology, Atlanta, Georgia, USA [2]Work partially done while an intern at JPMorgan [3]JPMorgan AI Research & AlgoCRYPT CoE, New York, New York, USA. Correspondence to: Alexander Bienstock <afb383@nyu.edu>, Yu Wei <ywei368@gatech.edu>.

*Proceedings of the 43rd International Conference on Machine Learning*, Seoul, South Korea. PMLR 306, 2026. Copyright 2026 by the author(s).

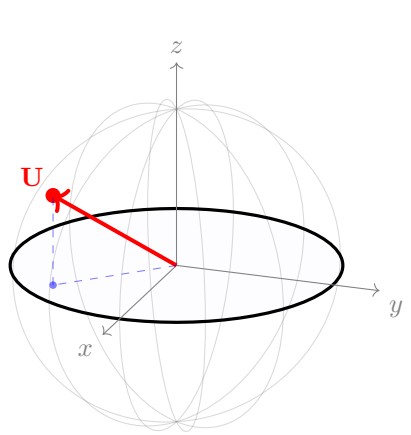
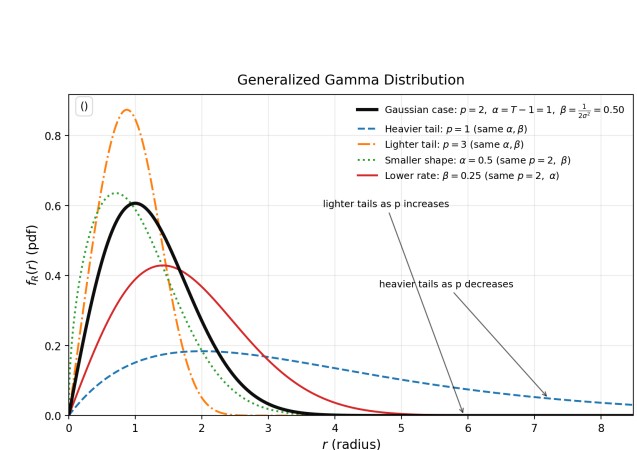

*Figure 1.* A Spherically Symmetric random variable $X = RU$ has independent (scalar) radial component $R$ and directional component $U$. **Left**: The directional component **U** is uniformly distributed on the unit sphere. **Right**: The density functions of one family of radial components $R$ that we study: the *Spherical Generalized Gamma* distribution with varying shape parameter $\alpha$, rate $\beta$, and tail control $p$.

et al., 2006a). Along the way, we provide additional insights, including answering an open question from (Joseph et al., 2025).

To answer (Q1), we first show that one only need focus on *spherical symmetric* noise distributions (see Figure 1); they perform at least as well as all other additive noise distributions. Indeed, spherical symmetric noise distributions satisfy rotational invariance and are thus a natural choice for the worst-case nature of DP, in which a single data record could perturb the query in any direction; both the Gaussian and $\ell_2$ mechanisms use spherical symmetric distributions. A bit more formally, spherical symmetric random variables are of the form $X = RU$, where $R \in \mathbb{R}_{\geq 0}$ is a radial random variable and $U \sim \mathsf{Unif}(\mathcal{S}^{T-1})$ is an independent random $T$-dimensional unit vector.

Now, consider the Gaussian privacy function $g_{\varepsilon,s}(\sigma^2)$, which for fixed privacy level $\varepsilon > 0$ and $\ell_2$ sensitivity $s > 0$, outputs the optimal $\delta_G$ of the Gaussian mechanism with per-coordinate variance $\sigma^2$ (Balle & Wang, 2018b; Lu et al., 2023). We show that for any fixed Mean Squared Error (MSE) constraint $e = \mathbf{Ex}\left[R^2\right]$ and any spherical noise $X = RU$, as $T \to \infty$, the optimal $\delta$ that $X$ can achieve for $(\varepsilon, s)$ is lower bounded, up to an $o(1)$ term, by $\mathbf{Ex}\left[g(R^2/T)\right]$, the average over $R$ of the Gaussian privacy function evaluated at the normalized radial energy $R^2/T$. This value $R^2/T$ for the Gaussian mechanism is exactly equal to the per-coordinate variance $\sigma^2 = e/T$ in the high-dimensional limit. Next, for large enough $e$, we show that $g$ satisfies certain properties that allow for a Jensen-like inequality

$$\mathbf{Ex}\left[g(R^2/T)\right] \geq g(\mathbf{Ex}\left[R^2/T\right]) = g(e/T) = \delta_G.$$

Thus, the Gaussian mechanism cannot be asymptotically improved upon in the sense of achieving a strictly smaller limiting worst-case optimal delta at the same MSE. We can thus deduce the following (since larger error yields stronger privacy):

**Theorem 1.1** (Informal version of Theorem 3.1)**.** *As the dimension of query $Q \in \mathbb{R}^T$, $T \to \infty$, using Gaussian noise $X$ in the additive noise mechanism $\widehat{Q} = Q + X$ yields lowest error among all noise distributions, for all strong enough privacy requirements.*

This result is of particular relevance to training deep learning models with DP (Abadi et al., 2016), in which $T$ is the number of model parameters, which can be quite large.

To answer (Q2), we consider for the first time a broad and expressive subfamily of spherical symmetric distributions that contains those of both the Gaussian and $\ell_2$ mechanisms, amongst many others: the *Spherical Generalized Gamma Distributions* (see right of Figure 1). We begin by demonstrating how to numerically evaluate the DP guarantees of these mechanisms. Indeed, by leveraging the radial decomposition perspective — where the spherical noise is separated into its magnitude and directional components — we reduce the privacy analysis to a one-dimensional integration over a smooth, absolutely continuous radial random variable. This structure allows the privacy loss to be evaluated efficiently and with arbitrary precision using standard numerical integration techniques. We furthermore bound the error that can arise from such numerical evaluation to obtain formal DP guarantees.

We leverage this ability in order to find settings and mechanisms for those settings that have less error than *both* the Gaussian and $\ell_2$ mechanisms. The spherical generalized gamma distribution is characterized by three parameters:

the shape parameters $\alpha$ and $p$, and the scale parameter $\beta$. We provide an optimization algorithm that finds the parameters which minimize the error of the corresponding additive noise mechanism, for a particular privacy level. Our evaluation shows that in low dimensions, we are able to find $\alpha, p, \beta$ that result in a mechanism with error up to 15% less than *both* the Gaussian and $\ell_2$ mechanisms.

As a bonus, we study tight composition of all Spherical Generalized Gamma Mechanisms using differential privacy accounting techniques. This demonstrates the compatibility of these mechanisms with existing DP frameworks and answers an open question regarding tight composition of the $\ell_2$ mechanism (Joseph et al., 2025).[1]

**Related Work.** Previous works have studied the optimality of the Gaussian mechanism: (Dong et al., 2021) show that a restricted class of additive noise mechanisms all converge to the Gaussian mechanism as $T \to \infty$ (in terms of Gaussian DP (Dong et al., 2022)), and that across all mechanisms in this class, the Gaussian mechanism performs best for *all* dimensions $T$. However, this class of additive noise mechanisms is quite restrictive, since we know, for example, that the $\ell_2$ mechanism outperforms the Gaussian mechanism for low $T$. Moreover, Gaussian DP is tailored towards the Gaussian mechanism and is a "relaxation" (Dong et al., 2022) of the classical approximate DP that we study.

Works that study the *non*-optimality of the Gaussian mechanism include (Geng & Viswanath, 2016), who only study *integer* valued (vector) queries in terms of $\ell_1$ sensitivity, and (Geng et al., 2020), who only study real valued *scalar* queries. The $\ell_2$ mechanism extends (Hardt & Talwar, 2010), which only studies pure DP, to approximate DP. Building off the family of Generalized Gaussian Mechanisms from (Liu, 2019), (Rinberg et al., 2025) provide *empirical* evidence that the Gaussian is optimal within this family. However, we are able to show notable improvements of non-Gaussian mechanisms in our Spherical Generalized Gamma Mechanism over the Gaussian mechanism. Moreover, Generalized Gaussian Mechanisms, specified by only *two* parameters, are a proper subset of our Spherical Generalized Gamma Mechanisms, specified by *three* parameters.

The Rank-1 Singular Multivariate Gaussian (R1SMG) mechanism introduced in (Ji & Li, 2024), is a special case within our broader family of spherical generalized gamma mechanisms. The authors claim that RS1MG has error an order of magnitude lower than the Gaussian mechanism; however, we discover a bug in their proof (see Appendix E). Indeed, our numerical evaluation reveals that the privacy guarantees of R1SMG are substantially weaker than those of the standard multivariate Gaussian mechanism.

---

[1] As per the review discussion: https://openreview.net/forum?id=ypeehAYK7W.

The work of (Alghamdi et al., 2023) studies the $(\varepsilon, \delta)$-DP composition of real valued vector queries. They show that as the number of sequential compositions, $k$, trends towards infinity, additive, spherical symetric mechanisms are optimal for $\ell_2$ sensitivity. They also provide an optimization algorithm to find the best such mechanism, but *only* in this asymptotic sense as $k \to \infty$. They provide some empirical evidence for small $k$ of their mechanism's superiority over the Gaussian mechanism for $T = 10$, but only for very small $\delta = 10^{-8}$ (note: it is already known that the Gaussian mechanism cannot perform well as $\delta \to 0$, and cannot achieve $\delta = 0$.). We on the other hand study the standalone (no composition) setting and show superior mechanisms to the Gaussian for more realistic values of $\delta$ like $\delta = 10^{-3}$.

## 2. Preliminaries

### 2.1. Differential Privacy

We consider a dataset space $\mathcal{X}^*$ equipped with a neighboring relation $\simeq$ (add/remove of one record). A randomized mechanism $\mathcal{M} : \mathcal{X}^* \to \mathbb{R}^T$ is $(\varepsilon, \delta)$-*differentially private* (Dwork & Roth, 2014) if for all neighboring $G \simeq G'$ and all set $\mathcal{S} \subseteq \mathbb{R}^T$,

$$\Pr\left[\mathcal{M}(G) \in \mathcal{S}\right] \le e^{\varepsilon} \Pr\left[\mathcal{M}(G') \in \mathcal{S}\right] + \delta \qquad (1)$$

**Hockey-stick divergence and optimal $\delta$.** Let $(t)_+ \stackrel{\text{def}}{=} \max\{t, 0\}$. For random variables $X, Y$ taking values on the same support $\mathbb{R}^T$ and $\varepsilon \ge 0$, define the *hockey-stick divergence* at level $\varepsilon$ (Balle et al., 2018)) (a.k.a., $\alpha$-divergence with $\alpha = e^{\varepsilon}$) by

$$\mathsf{H}_{\varepsilon}\left(X, Y\right) \stackrel{\text{def}}{=} \sup_{S \in \mathbb{R}^T} \left(\Pr[X \in S] - e^{\varepsilon} \Pr[Y \in S]\right)_+,$$

In the context of differential privacy, for a fixed neighboring pair $G \simeq G'$, we interpret $\mathsf{H}_{\varepsilon}\left(\mathcal{M}(G), \mathcal{M}(G')\right)$ as the *optimal (minimal) $\delta$* for the (one-sided) DP inequality $\Pr\left[\mathcal{M}(G) \in \mathcal{S}\right] \le e^{\varepsilon} \Pr\left[\mathcal{M}(G') \in \mathcal{S}\right] + \delta$ holding for all measurable $S$. Accordingly, we define the *(one-side) optimal delta* (Lu et al., 2024) at privacy level $\varepsilon$

$$\delta_{G,G'}(\varepsilon) \stackrel{\text{def}}{=} \mathsf{H}_{\varepsilon}\left(\mathcal{M}(G), \mathcal{M}(G')\right) \qquad (2)$$

and the *(worst-case) optimal delta* (a.k.a., privacy profile (Balle et al., 2018)) of the mechanism at privacy level $\varepsilon$ as

$$\delta_{\mathcal{M}}(\varepsilon) \stackrel{\text{def}}{=} \sup_{G \simeq G'} \delta_{G,G'}(\varepsilon). \qquad (3)$$

By construction, $\mathcal{M}$ is $(\varepsilon, \delta)$-DP if and only if $\delta_{\mathcal{M}}(\varepsilon) \le \delta$.

| Radial r.v. $R$ | Radial density $f_R(r)$ | Noise density $f_X(x) = g(\|x\|_2^2)$ (with $y = \|x\|_2$) | Shape $f_X(x) \propto$ |
|---|---|---|---|
| GGamma$(\alpha, \beta, p)$ | $\dfrac{p\beta^{(\alpha+1)/p}}{\Gamma(\frac{\alpha+1}{p})} r^\alpha e^{-\beta r^p}$ | $\dfrac{\Gamma(T/2)}{2\pi^{T/2}} \dfrac{p\beta^{(\alpha+1)/p}}{\Gamma(\frac{\alpha+1}{p})} y^{\alpha+1-T} e^{-\beta y^p}$ | $y^{\alpha+1-T} e^{-\beta y^p}$ |
| GGamma$\left(T-1, \frac{1}{2\sigma^2}, 2\right)$ (Gaussian in $\mathbb{R}^T$) | $\dfrac{r^{T-1}}{2^{\frac{T}{2}-1}\Gamma(\frac{T}{2})\sigma^T} e^{-\frac{r^2}{2\sigma^2}}$ | $\dfrac{1}{(2\pi\sigma^2)^{T/2}} e^{-\frac{y^2}{2\sigma^2}}$ | $e^{-\frac{y^2}{2\sigma^2}}$ |
| GGamma$\left(T-1, \frac{1}{\theta}, 1\right)$ ($\ell_2$-Laplace in $\mathbb{R}^T$) | $\dfrac{r^{T-1}}{\theta^T \Gamma(T)} e^{-\frac{r}{\theta}}$ | $\dfrac{\Gamma(T/2)}{2\pi^{T/2}\theta^T\Gamma(T)} e^{-\frac{y}{\theta}}$ | $e^{-\frac{y}{\theta}}$ |

*Table 1.* Radial random variables $R$ and the induced spherically symmetric densities $f_X(x) = g(\|x\|_2^2)$ for noise random variable $X = RU$, obtained via Theorem 2.1. The last column reports $g(\|x\|_2^2)$ up to a normalization constant.

## 2.2. Gaussian Mechanism

We consider the $T$-dimensional vector-valued query $q : \mathcal{X}^* \to \mathbb{R}^T$ with $\ell_2$ sensitivity

$$s \stackrel{\text{def}}{=} \sup_{G \simeq G'} \|q(G) - q(G')\|_2.$$

For $u > 0$, the Gaussian mechanism (Dwork et al., 2006b; Balle & Wang, 2018b) with per-coordinate variance $u$ is

$$\mathcal{M}_{\mathsf{G},u}(G) \stackrel{\text{def}}{=} q(G) + G, \qquad G \sim \mathcal{N}(0, uI_T). \quad (4)$$

Notice that the mechanism's mean-squared error is

$$\max_{G \in \mathcal{X}^*} \mathbf{Ex}\left[\|\mathcal{M}_{\mathsf{G},u}(G) - q(G)\|_2^2\right] = \mathbf{Ex}\left[\|G\|_2^2\right] = Tu. \quad (5)$$

Denote by $\Phi(\cdot)$ the standard normal CDF. According to the literature (Balle & Wang, 2018a; Lu et al., 2023), the optimal delta of $\mathcal{M}_{\mathsf{G},u}$ at privacy level $\varepsilon$ is

$$\delta_{\mathcal{M}_{\mathsf{G},u}}(\varepsilon) = g(u)$$
$$\stackrel{\text{def}}{=} \Phi\left(-\frac{\varepsilon\sqrt{u}}{s} + \frac{s}{2\sqrt{u}}\right) - e^\varepsilon \Phi\left(-\frac{\varepsilon\sqrt{u}}{s} - \frac{s}{2\sqrt{u}}\right). \quad (6)$$

In particular, for fixed $(\varepsilon, \delta, s)$, the minimal Gaussian variance achieving $(\varepsilon, \delta)$-DP is

$$u_0(\delta) \stackrel{\text{def}}{=} \inf\{u > 0 : g(u) \leq \delta\}. \quad (7)$$

We will repeatedly use Equation (6) as the baseline privacy–utility tradeoff to compare against other additive-noise mechanisms.

## 2.3. Spherical Symmetric Distribution

We begin by defining *spherically symmetric* random vectors $X \in \mathbb{R}^T$, which are the main noise distributions studied in this paper.

**Definition 2.1** (Spherically Symmetric Distribution (Fang, 2018) [2]). A random vector $X \in \mathbb{R}^T$ is *spherically symmetric* if there exists a random variable $R \geq 0$ and a random

---

[2] The first characterization can be found in Thm 2.1 and the second appears in equation 2.7 of the book (Fang, 2018)

---

vector $U$ uniformly distributed on the unit sphere $\mathcal{S}_{++}^T$ such that $X$ can be decomposed as:

$$X = RU,$$

where $R$ and $U$ are independent. Equivalently, $X$ is *spherically symmetric* if its density function $f_X(x)$ exists and depends only on the norm of $x$, that is,

$$f_X(x) = g(\|x\|_2^2),$$

for some deterministic function $g : [0, \infty) \to \mathbb{R}_{\geq 0}$.

The decomposition $X = RU$ separates *radial* ($R$) and *direction* ($U$) random variables. Theorem 2.1 shows how to convert a radial density into the induced spherically symmetric density. Table 1 lists some spherical random variables and the corresponding radial random variables studied in the paper.

**Theorem 2.1** (Radial-to-Spherical Density Transformation (Fang, 2018)[Thm 2.9]). *Let $X = RU$ be a spherically symmetric random variable. If the random variable $R$ has a density over $\mathbb{R}_{\geq 0}$ and denote its pdf as $f_R(\cdot)$, then,*

$$f_X(x) = g(\|x\|_2^2) = \frac{\Gamma(\frac{T}{2})}{2\pi^{\frac{T}{2}}} \|x\|_2^{1-T} f_R(\|x\|_2).$$

In Appendix A, we give Proposition A.1 that characterizes the distribution of $\cos\Theta$, where $\Theta$ is the angle between the random direction vector $U$ and a fixed reference direction.

## 3. Gaussian Optimality in High Dimensions

We study the optimality of the Gaussian noise distribution for additive-noise differentially private mechanisms in high dimensions. A natural utility measure for this class is the mean-squared error (MSE), equivalently, the second moment of the additive noise (with zero-centered mean). Our notion of optimality asks which mechanism provides the strongest privacy guarantee under a fixed MSE budget. Concretely, we compare mechanisms at a fixed privacy level $\varepsilon$ and sensitivity $s$ by their *optimal delta* $\delta_{\mathcal{M}}(\varepsilon)$ (cf. Equation (3)); smaller $\delta_{\mathcal{M}}(\varepsilon)$ yields stronger privacy.

Theorem 3.1 states our main optimality result. Informally, for a fixed privacy level $\varepsilon$ and sensitivity $s$ and a target high-privacy regime with $\delta$ sufficiently small, Theorem 3.1 says that, as the query dimension $T \to \infty$, the Gaussian mechanism is asymptotically optimal among all additive-noise mechanisms with the same MSE: if an additive-noise mechanism matches the Gaussian error budget, then its optimal delta at privacy level $\varepsilon$ cannot be smaller than $\delta$.

**Theorem 3.1** (Asymptotic Gaussian optimality among additive noises). *Fix $\varepsilon \geq 0$ and $s > 0$. There exists $\delta_\star \in (0, 1)$ such that for all $\delta \in (0, \delta_\star]$ the following holds.*

*Let $u_0 = u_0(\delta)$ be the minimal Gaussian variance for which the Gaussian mechanism $\mathcal{M}_{\mathsf{G}, u_0}$ is $(\varepsilon, \delta)$-DP. For each $T \geq 2$, let $\mathcal{M}_{T, u_0}(\mu_T) \stackrel{\text{def}}{=} \mu_T + X_T$ be any additive-noise mechanism with $\mathbf{Ex}\left[\|X_T\|_2^2\right] = Tu_0$. Then,*

$$\liminf_{T \to \infty} \delta_{\mathcal{M}_{T, u_0}}(\varepsilon) \geq \delta_{\mathcal{M}_{\mathsf{G}, u_0}}(\varepsilon) = \delta.$$

*Equivalently, for every $\eta > 0$, for all sufficiently large $T$,*

$$\delta_{\mathcal{M}_{T, u_0}}(\varepsilon) \geq \delta - \eta.$$

**Numerical lower bounds on the threshold $\delta_\star$.** Theorem 3.1 is stated in the privacy regime $\delta \leq \delta_\star$, where $\delta_\star \in (0, 1)$ is a constant. To demonstrate that this regime is non-vacuous and, in particular, includes $\delta$-values typically used in practice, Table 2 reports numerically verified lower bounds on $\delta_\star$ for some $\varepsilon$ (with sensitivity fixed to $s = 1$). The values in Table 2 provide evidence that the privacy regime $\delta \leq \delta_\star$ covers standard privacy choices such as $\delta \leq 10^{-3}$ (in fact, the range is much bigger), thereby highlighting the practical relevance of Theorem 3.1. Our computation is based on a numerical verification of the tangent-support conditions for the Gaussian privacy function $g$, explained below (cf. Proposition 3.1).

**Finite-dimensional convergence.** Theorem 3.1 is an asymptotic statement in the dimension $T$. In particular, it does not provide a concrete threshold $T_0$ or a convergence rate for how quickly the Gaussian's asymptotic optimality kicks in. The only asymptotic loss in the proof enters through Lemma 3.1, where the distribution of a suitably scaled coordinate of a uniform random direction is approximated by a standard normal distribution. Making the result quantitative would require replacing this $o(1)$ aproximation gap by an explicit normal-approximation bound and propagating that error through the distinguishing test used in the proof. We leave such finite-$T$ rates as an interesting direction for future work.

## 3.1. Proof sketch of Theorem 3.1

In this section, we provide a proof sketch of Theorem 3.1. Full details are deferred to Appendix B.

*Table 2.* Numerical lower bounds on $\delta_\star$ in Theorem 3.1 (for some $\varepsilon$ with fixed sensitivity $s = 1$).

| $\varepsilon$ | $\delta_\star$ | $\varepsilon$ | $\delta_\star$ |
|---|---|---|---|
| 0.25 | 0.736670 | 4.00 | 0.416972 |
| 0.50 | 0.706970 | 8.00 | 0.292170 |
| 1.00 | 0.649185 | 16.00 | 0.197615 |
| 2.00 | 0.549133 | | |

We first focus our discussion on *spherically symmetric* noises, a natural yet broad class in the DP worst-case model: since neighboring datasets can induce sensitivity in arbitrary directions, it is natural to consider perturbations that treat all directions uniformly. We define the class of additive-noise mechanism adding *spherically symmetric* noises as *spherical additive-noise mechanisms*, and formally state it in Definition 3.1.

**Definition 3.1** (Spherical additive-noise mechanism). Fix $T \geq 2$ and an $\ell_2$-sensitive query with sensitivity $s > 0$ under the add/remove neighboring relation. A *spherical additive-noise mechanism* is any mechanism of the form

$$\mathcal{M}_{R_T, T}(\mu_T) \stackrel{\text{def}}{=} \mu_T + R_T U_T,$$

where $\mu_T \in \mathbb{R}^T$ is the dataset query result, $U_T \sim \mathsf{Unif}(\mathcal{S}^{T-1})$ is uniform on the unit sphere, and $R_T \in \mathbb{R}_{>0}$ is independent of $U_T$.

Lemma 3.1 states our first observation that, when the query dimension $T$ is sufficiently large, the optimal $\delta$ of any spherical additive-noise mechanism $\mathcal{M}_{R_T, T, u_0}$ can be lower bound, up to an asymptotically vanishing error term, in terms of the Gaussian privacy function $g(\cdot)$ (cf. Equation (6)). Specifically, letting $R_T$ denote the radial random variable of the spherical noise, the optimal $\delta$ at privacy level $\varepsilon$ is lower bounded by $\mathbf{Ex}\left[g\left(\frac{R_T^2}{T}\right)\right]$, which is the Gaussian privacy expression evaluated at the *normalized radial energy* $R_T^2/T$ and averaged over its randomness. This reduction turns the comparison between the Gaussian mechanism and an arbitrary spherical mechanism into a one-dimensional question: under the mean-preserving constraint $\mathbf{Ex}\left[R_T^2/T\right] = u_0$ (the MSE budget), can any randomization $R_T^2/T$ make $\mathbf{Ex}\left[g(R_T^2/T)\right]$ smaller than $g(\mathbf{Ex}\left[R_T^2/T\right]) = g(u_0)$? We note that for Gaussian benchmark, its corresponding $R_T^2/T$ equals to $u_0 \cdot \chi_T^2/T$, which converges to $u_0$ as $T \to \infty$.

**Lemma 3.1** (Proof is in Appendix B). *Fix $\varepsilon \geq 0$ and $s > 0$. For each $T \geq 2$, let $U_T \sim \mathsf{Unif}(\mathcal{S}^{T-1})$ and let $R_T \in \mathbb{R}_{>0}$ be independent of $U_T$. For any deterministic $\mu_T \in \mathbb{R}^T$ with $\|\mu_T\|_2 = s$, define $X_T = R_T U_T$ and $Y_T = X_T + \mu_T$. Let $\delta$ be the optimal delta with respect to $X_T, Y_T$ at privacy*

*level $\varepsilon$. Then, as $T \to \infty$, it holds that*

$$\delta \geq \mathbf{Ex}\left[g\left(\frac{R_T^2}{T}\right)\right] + o(1).$$

Intuitively, to prove Lemma 3.1, we lower bound the optimal $\delta$ between $X_T$ and $Y_T$ by constructing a set $\mathcal{S}_T$ and bounding $\Pr[X \in \mathcal{S}_T] - e^{\varepsilon}\Pr[Y \in \mathcal{S}_T]$ from below. We choose $\mathcal{S}_T$ to be a threshold region — an affine test with a mild radius-dependent correction — designed to mirror the affine test for distinguishing Gaussian shifts. Conditioning on $R_T = r$, we analyze the membership probabilities $\Pr[X_T \in \mathcal{S}_T \mid R_T = r]$ and $\Pr[Y_T \in \mathcal{S}_T \mid R_T = r]$, and show that as $T \to \infty$, they converge to the two Gaussian CDF terms that define the Gaussian privacy function evaluated at $u = r^2/T$. Averaging over $R_T$ then yields the stated asymptotic lower bound in Lemma 3.1.

Proposition 3.1 answers the one-dimensional question induced by Lemma 3.1: if the Gaussian privacy function $g(\cdot)$ satisfies an appropriate supporting-line property at the MSE budget $u_0$, then any mean-preserving randomization of the effective variance cannot improve privacy relative to the Gaussian benchmark. Concretely, the supporting-line property requires that $g$ is convex on $[u_0, \infty)$ and that $g$ lies above its tangent line at $u_0$ on $[0, u_0]$. This is a tailored Jensen-type condition without requiring global convexity: convexity on the upper side enables a Jensen argument, while the additional lower-side tangent bound ensures the same supporting line remains valid even where $g$ need not be convex. A formal proof appears in Appendix B.

**Proposition 3.1** (Proof is in Appendix B). *Let $\varepsilon \geq 0$, $\delta \in (0,1)$ and $s > 0$, and recall the Gaussian privacy function $g(\cdot)$ defined in Equation (6). Let $u_0 > 0$ be the unique point such that $g(u_0) = \delta$ (i.e., the minimal Gaussian variance shown in Equation (7)). Define the tangent line function at $u_0$ by*

$$\ell_\delta(u) \stackrel{\text{def}}{=} g(u_0) + g'(u_0)(u - u_0). \tag{8}$$

*If $g$ is convex on $[u_0, \infty)$ and $g(u) \geq \ell(u)$ for all $u \in [0, u_0]$, then for every random variable $U \geq 0$ with $\mathbf{Ex}[U] = u_0$, $\mathbf{Ex}[g(U)] \geq g(u_0)$.*

Lemma 3.2 asserts that, in the high-privacy regime (i.e., for sufficiently small $\delta$), the Gaussian privacy function $g(\cdot)$ satisfies the supporting-line property required by Proposition 3.1. Concretely, for all $\delta$ in this regime, the corresponding minimal MSE level $u_0(\delta)$ lies in a well-behaved region of $g$: the function is convex on $[u_0(\delta), \infty)$ and, moreover, the tangent line at $u_0(\delta)$ lower bounds $g$ throughout the interval $[0, u_0(\delta)]$.

**Lemma 3.2** (Proof is in Appendix B). *Let $\varepsilon \geq 0$, $s > 0$, and recall the Gaussian privacy function $g(\cdot)$ defined in*

*Equation (6). For each $\delta \in (0,1)$, let $u_0(\delta) > 0$ be the unique point such that $g(u_0(\delta)) = \delta$ (i.e., the minimal Gaussian variance shown in Equation (7)). Denote the tangent line function at $u_0(\delta)$ by $\ell_\delta(u)$ (cf. Equation (8)).*

*Then there exists $\delta_\star \in (0,1)$, such that for every $\delta \in (0, \delta_\star]$ the following hold:*

1. *$g$ is convex on $[u_0(\delta), \infty)$;*
2. *$g(u) \geq \ell_\delta(u)$ for all $u \in [0, u_0(\delta)]$.*

Intuitively, to prove Lemma 3.2, our first step is an asymptotic analysis of $g(u)$ as the MSE budget $u \to \infty$ (equivalently, $\delta = g(u) \to 0$): using Mills-type bounds for $\Phi$, we obtain the expansion

$$g(u) = C_{\varepsilon,s} u^{-3/2} \exp\left(-\frac{\varepsilon^2}{2s^2}u\right)\left(1 + O\left(\frac{1}{u}\right)\right),$$

which implies that $g$ is eventually convex, i.e., there exists $u_{\text{right}}$ such that $g$ is convex on $[u_{\text{right}}, \infty)$. To control the lower part, we study the tangent-line intercept $L(u) = g(u) - ug'(u)$. The same asymptotics show that $L(u) \to 0$ as $u \to \infty$. Since $g$ is decreasing, we can choose a sufficiently large point $u_\star \geq u_{\text{right}}$ so that the tangent line at $u_\star$ has intercept below $\min_{u \in [0, u_{\text{right}}]} g(u)$; this ensures the tangent line at $u_\star$ lies below $g$ on $[0, u_{\text{right}}]$, while convexity guarantees the supporting-line inequality on $[u_{\text{right}}, u_\star]$. Defining $\delta_\star \stackrel{\text{def}}{=} g(u_\star)$ yields the desired supporting-line property at $(u_\star, \delta_\star)$. Finally, we use the downward-closure argument (Proposition B.1) to extend the property from $\delta_\star$ to all $\delta \leq \delta_\star$.

Combining Lemmas 3.1 and 3.2 and proposition 3.1 yields the conclusion that, in the high-privacy regime, the Gaussian mechanism is asymptotically optimal among all *spherical* additive-noise mechanisms with the same MSE. To extend this optimality beyond spherical noises, Lemma 3.3 formalizes the last piece. The idea is to reduce an arbitrary additive noise $X$ to a spherically symmetric one $X'$ via Haar symmetrization.

Lemma 3.3 says that the symmetrization preserves the MSE and cannot increase the worst-direction optimal delta over shifts of $\ell_2$ norm at most $s$. The proof uses convexity of hockey-stick divergence under mixtures together with orthogonal invariance. Therefore, under a fixed MSE budget, the optimal privacy achievable by additive-noise mechanisms is no better than that achievable by spherical additive-noise mechanisms.

**Lemma 3.3** (Haar symmetrization (Proof is in Appendix B)). *Fix $T \geq 2$, $\varepsilon \geq 0$, and $s > 0$. Let $M$ be a Haar-uniform random matrix over the group of all $T \times T$ orthogonal matrices and independent of a random vector $X \in \mathbb{R}^T$. Define the symmetrized noise $X' \stackrel{\text{def}}{=} MX$. Let $\delta \stackrel{\text{def}}{=} \sup_{\|v\|_2 \leq s} \mathsf{H}_\varepsilon(X, X + v)$ and $\delta' \stackrel{\text{def}}{=}$*

$\sup_{\|v\|_2 \leq s} \mathsf{H}_\varepsilon (X', X' + v)$ *denote the worst-direction optimal deltas at privacy level* $\varepsilon$ *over the* $\ell_2$ *ball of radius s (cf. Equation* (2)*).*

*Then* $X'$ *is spherically symmetric, and*

$$\mathbf{Ex}\left[\|X'\|_2^2\right] = \mathbf{Ex}\left[\|X\|_2^2\right] \qquad \delta' \leq \delta.$$

Lemmas 3.1 to 3.3 and proposition 3.1 yields Theorem 3.1 immediately. We defer a formal wrap-up to Appendix B.

# 4. Spherical Generalized Gamma Mechanism

In this section, we study the question of additive-noise design in finite (especially low) dimensions, where the *shape* of the noise can materially affect privacy–utility trade-off. Concretely, we formalize the Spherical Generalized Gamma (SGG) noise family and its use as an additive DP primitive for $\ell_2$-sensitive vector queries.

Definition 4.1 defines the three-parameter generalized-gamma family, denoted by $R \sim \mathsf{GGamma}(\alpha, \beta, p)$, which we use to model the *radial* component of our noise. Intuitively, $\alpha$ controls the near-zero behavior (shape), $\beta$ sets the overall scale (noise magnitude), and $p$ tunes tail decay; see Figure 1 (right) for a visual.

**Definition 4.1.** A three-parameter **generalized-Gamma** $R \sim \mathsf{GGamma}(\alpha, \beta, p)$ is defined with the density function:

$$f_{\alpha,\beta,p}(r) = \frac{p\,\beta^{(\alpha+1)/p}}{\Gamma\left(\frac{\alpha+1}{p}\right)} r^\alpha \exp\left(-\beta r^p\right),\ \alpha > -1,\ p > 0,\ \beta > 0.$$

Definition 4.2 defines the *spherical generalized-gamma* (SGG) family, induced by a generalized-gamma *radial* distribution. The SGG family includes several standard spherical noises. For instance: (i) setting $(p, \alpha) = (2, T-1)$ and $\beta = 1/(2\sigma^2)$ yields $X \sim \mathcal{N}\left(0, \sigma^2 I_T\right)$ (Gaussian noise); (ii) setting $(p, \alpha) = (1, T-1)$ and $\beta = 1/\theta$ yields spherical (i.e., $\ell_2$-) Laplace noise with scale $\theta$.

**Definition 4.2** (Spherical generalized-gamma (SGG) noise)**.** Fix a dimension $T \geq 2$ and parameters $\alpha > -1$, $\beta > 0$, $p > 0$. A random vector $X \in \mathbb{R}^T$ follows a *Spherical Generalized-Gamma (SGG)* distribution, denoted by $X \sim \mathsf{SGG}(\alpha, \beta, p)$, if $X = RU$, where $R \sim \mathsf{GGamma}(\alpha, \beta, p)$ and $U \sim \mathsf{Unif}(\mathcal{S}^{T-1})$ are independent.

## 4.1. Privacy Analysis

Lemma 4.1 shows that the optimal delta at privacy level $\varepsilon$ between SGG noise $X$ and its shift $X + \mu$ admits an explicit *one-dimensional* integral representation. In particular, the optimal $\delta_\mu^*(\varepsilon)$ can be evaluated by integrating a smooth integrand over $r \in (0, \infty)$, which yields an efficient and numerically stable procedure (in contrast to a direct exponentially expensive computation over $\mathbb{R}^T$).

**Lemma 4.1** (Proof is in Appendix C.1)**.** *Fix* $T \geq 2$ *and parameters* $\alpha \in (-1, T-1]$, $\beta > 0$, *and* $p > 0$. *Let* $X = RU \sim \mathsf{SGG}(\alpha, \beta, p)$ *and let* $\mu \in \mathbb{R}^T$ *be a shift vector with* $s = \|\mu\|_2$. *Let* $W = \cos \Theta = \frac{\langle \mu, U \rangle}{\|\mu\|_2} \in [-1, 1]$, *with CDF* $F_W$. *Then the optimal* $\delta_\mu^*(\varepsilon)$ *for the neighboring pair* $(X, X + \mu)$ *satisfies*

$$\delta_\mu^*(\varepsilon) = \max\Big\{0, \int_0^\infty f_R(r)\left(1 - F_W(w^*(r, -\varepsilon))\right) dr$$

$$- e^\varepsilon \int_0^\infty f_R(r)\, F_W(w^*(r, \varepsilon))\, dr\Big\},$$

*where* $f_R$ *is the density of* $R \sim \mathsf{GGamma}(\alpha, \beta, p)$, *and for each* $y \in \mathbb{R}$ *and* $r > 0$, $w^*(r, y) \in [-1, 1]$ *denotes the unique solution in* $w$ *to*

$$y =$$
$$\frac{\alpha + 1 - T}{2} \ln\Big(1 + \frac{2sw}{r} + \frac{s^2}{r^2}\Big) + \beta\Big[r^p - (r^2 + 2swr + s^2)^{p/2}\Big].$$

Informally, the proof follows by observing that under the radial–directional decomposition $X = RU$, the privacy loss can be expressed as a function of two *independent* one-dimensional random variables: the radial component $R$ and the directional statistic $W$ (the cosine of the angle between $U$ and $\mu$). Moreover, monotonicity in $W$ converts the resulting two-dimensional inequality involving $(R, W)$ into a one-dimensional threshold condition on $W$ given a fixed $R$, allowing us to integrate over $R$ and obtain the claimed one-dimensional representation.

Proposition 4.1 shows that for $\alpha \leq T - 1$ the SGG density is non-increasing with the radius: points farther from the origin receive no larger probability density.[3]

**Proposition 4.1** (Proof is in Appendix C.1)**.** *Fix a dimension* $T \geq 2$ *and parameters* $\alpha \in (-1, T-1]$, $\beta \geq 0$, $p > 0$. *Let* $X \sim \mathsf{SGG}(\alpha, \beta, p)$, *and let* $g : [0, \infty) \to [0, \infty)$ *denote the density generator such that* $f_X(x) = g(\|x\|_2^2)$. *Then* $y \mapsto g(y^2)$ *is non-increasing for* $y > 0$.

Building on this, Lemma 4.2 shows that, for the neighboring pair $(X, X + \mu)$, the optimal delta $\delta_\mu^*(\varepsilon)$ is monotone in the shift magnitude $\|\mu\|_2$. In particular, larger shifts are easier to distinguish and hence induce larger divergence, so the worst-case divergence over all shifts is attained at the largest feasible $\|\mu\|_2$.

**Lemma 4.2** (Proof is in Appendix C.1)**.** *Fix* $T \geq 2$ *and parameters* $\alpha \in (-1, T-1]$, $\beta > 0$, *and* $p > 0$, *and let* $X \sim \mathsf{SGG}(\alpha, \beta, p)$. *For any shift vector* $\mu \in \mathbb{R}^T$, *let* $\delta_\mu^*(\varepsilon)$ *denote the optimal* $\delta$ *for the neighboring pair* $(X, X + \mu)$ *at point* $\varepsilon$. *Then for any* $\mu_1, \mu_2 \in \mathbb{R}^T$ *with* $\|\mu_1\|_2 \leq \|\mu_2\|_2$ *and any* $\varepsilon > 0$, $\delta_{\mu_1}^*(\varepsilon) \leq \delta_{\mu_2}^*(\varepsilon)$.

---

[3] We note this concerns $f_X(x)$ over $\mathbb{R}^T$, not the radial density $f_R(r)$: due to the surface-area factor $r^{-(T-1)}$ in $f_X(x)$, $f_R(r)$ need not be decreasing even if $f_X(x)$ decreases with $r = \|x\|_2$.

Lemma 4.1, together with the monotonicity result in Lemma 4.2, indicates that calibrating an additive SGG mechanism for $\ell_2$-sensitive real-vector queries is, a numerical task: given parameters $(\alpha, p)$ and sensitivity $s$, one can evaluate $\delta_s^*(\varepsilon)$ as a function of the noise multiplier $\beta$, and then choose $\beta$ to meet a target $(\varepsilon, \delta)$ guarantee.

However, a naïve implementation of Lemma 4.1 may incur numerical errors (e.g., from truncating an improper integral). Since our goal is to have a *provable* differentially private primitive, we abstract these numerical subtleties through the oracle interface in Definition 4.3. Concretely, given $(\alpha, p, \beta, \varepsilon)$ and a shift magnitude $s$, the oracle outputs an $\eta$-tight *upper bound* on $\delta_\mu^*(\varepsilon)$, for all shifts $\mu$ with $\|\mu\|_2 \leq s$. Lemma 4.3 says that such an oracle exists.

**Definition 4.3.** Fix a dimension $T \geq 2$ and parameters $\alpha \in (-1, T-1]$, $\beta > 0$, $p > 0$, and $\varepsilon \geq 0$. Let $X \sim \mathsf{SGG}(\alpha, \beta, p)$. Fix an $\ell_2$-sensitivity bound $s \geq 0$ and a target slack $\eta \geq 0$. An algorithm $\mathsf{Oracle}_{\mathsf{SGG}}$ : $(\alpha, p, \beta, \varepsilon, s) \mapsto \widehat{\delta}$ is called an $\eta$-*tight upper-bound oracle* (for $\delta_\mu^*(\varepsilon)$) if, for every $\varepsilon \geq 0$ and every $\mu \in \mathbb{R}^T$ with $\|\mu\|_2 \leq s$, it outputs $\widehat{\delta}$ satisfying

$$\delta_\mu^*(\varepsilon) \leq \mathsf{Oracle}_{\mathsf{SGG}}(\alpha, p, \beta, \varepsilon, \|\mu\|_2) \leq \delta_\mu^*(\varepsilon) + \eta.$$

**Lemma 4.3** (Proof is in Appendix C.2)**.** *There exists a (determinstic) $\eta$-tight upper-bound oracle.*

Lemma 4.3 follows by an explicit construction. The oracle reduces the one-dimensional integrals defining $\delta_\mu^*(\varepsilon)$ to expectations under a standard $\Gamma(k, 1)$ measure via the change of variables $Z = \beta R^p$ (with $k = (\alpha + 1)/p$), so that Gamma tail probabilities can be computed in closed form. It then (i) truncates the integral at a data-independent threshold $z_{\max}$ chosen to make the discarded tail contribute at most $\eta_{\mathsf{tail}}$, and (ii) estimates the remaining trucated expectations using a standard binning scheme.

We then in Appendix C.2 provide a parameterized differentially private mechanism $\mathcal{M}_{\mathsf{SGG}}^{\alpha, p}$ (cf. Algorithm 4) for vector-valued queries. For fixed $(\alpha, p)$, the mechanism uses an intuitive bracketing-and-bisection search via $\mathsf{Oracle}_{\mathsf{SGG}}$ to calibrate the noise: it finds (up to tolerance $\tau$) the largest feasible rate parameter $\beta$ — equivalently, the smallest noise magnitude under the parameterization — subject to the target $(\varepsilon, \delta)$ constraint, and then outputs the noisy answer $q(G) + X$ with $X \sim \mathsf{SGG}(\alpha, \beta, p)$.

### 4.2. Utility-Optimized SGG Mechanism

Algorithm 4 shows that for any fixed $(\alpha, p)$, we can calibrate the rate parameter $\beta$ so that $\mathcal{M}_{\mathsf{SGG}}^{\alpha, p}$ satisfies the target $(\varepsilon, \delta)$ guarantee. However, the same privacy level can typically be achieved by many different pairs $(\alpha, p)$, and these choices can lead to different accuracy. This motivates selecting $(\alpha, p)$ *to minimize a prescribed error metric* (MSE)

while preserving the same privacy constraint.

#### 4.2.1. FINDING OPTIMAL PARAMETERS

We next provide Algorithm 5 (c.f. Appendix C.2), that numerically tunes the Spherical Generalized Gamma Mechanism by searching for parameters $(\alpha, \beta, p)$ that minimize the mean-squared error (MSE) $c$ subject to a target $(\varepsilon, \delta)$-privacy constraint. It proceeds by running a binary search over MSE values $c$. At each binary-search step, it solves the two-dimensional problem of selecting $\alpha^*, p^*$ to minimize the achieved optimal delta $\delta^\star$ under the fixed noise budget $c$. For each candidate $(\alpha, p)$, Algorithm 4 determines whether there exists a noise multiplier $\beta^\star$ that satisfies the MSE constraint $c$ and returns the corresponding $\delta^\star$; we implement the $(\alpha, p)$ search using standard Bayesian optimization. If $\delta^* \leq \delta$, the target parameter, then the noise is decreased in the next iteration; otherwise it is increased. The algorithm uses in the mechanism output the parameters $\alpha^*, \beta^*, p^*$ for the distribution of radial component $R$ that minimize the MSE $c$ subject to $\delta^* \leq \delta$.

### 4.3. Advantage over Gaussian and $\ell_2$ Mechanisms

Based on Algorithm 5, we use the SGG family as a search space to identify selected low-dimensional regimes where an SGG mechanism outperforms both the Gaussian and $\ell_2$ mechanisms. The purpose of this experiment is to demonstrate that there is a low dimensional regimens that we could have improvement over both gaussian and $\ell_2$; it is not intended to show that SGG uniformly or systematically dominates Gaussian and $\ell_2$ across privacy parameters or dimensions.

We know that the $\ell_2$ mechanism (high-dimensional version of Laplace) is best suited for $\delta_{\ell_2} \to 0$, while the Gaussian mechanism is suited for $\delta_G > 0$. Thus, we hypothesize that the SGG Mechanism will have the best advantage at some $\delta^*$ that is not too small or too big, for fixed privacy parameter $\varepsilon > 0$, dimension $T$, and sensitivity $s$. Based on this intuition, we develop Algorithm 6 (in Appendix C.4) that takes in $\varepsilon, T, s$ and performs a binary search (using Algorithm 5) to find the $\delta^*$ at which the MSE of the SGG mechanism has the biggest advantage compared to the smaller of the MSEs of the Gaussian and $\ell_2$ mechanisms.

Using Algorithm 6, we identify realistic settings in which the optimal SGG distribution is much different from the Gaussian and $\ell_2$ mechanisms ($p$ far from 1 and 2), and has MSE up to 15% lower than them. In Figure 2 we demonstrate our findings. Unsurprisingly given Theorem 3.1, the improvements are larger for small dimension $T$ and get smaller as $T$ increases.

**Scope of the low-dimensional improvement.** The gains in Figure 2 should be interpreted as a principled existence

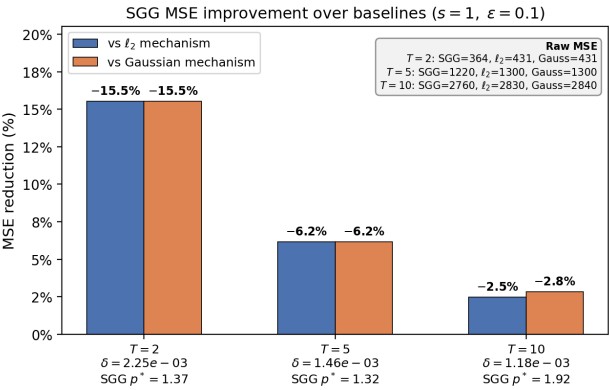

*Figure 2.* MSE reduction of the SGG mechanism over the $\ell_2$ and Gaussian baselines with $\varepsilon = 0.1$ and sensitivity $s = 1$. Each group shows the dimension $T$, the $\delta$ at which this advantage occurs, and the optimal SGG shape parameter $p^*$.

result rather than as a uniform recommendation to replace the Gaussian mechanism in concrete dimensional mechanism design. Our study does not show strong evidence that non-Gaussian SGG mechanisms consistently yield significant improvements over Gaussian across all dimensions and privacy parameters. In our experiments, the advantage is concentrated in selected low-dimensional regimes and shrinks rapidly as the dimension increases. Thus, SGG is best viewed as a useful design space for finite-dimensional additive-noise mechanisms, while the Gaussian mechanism remains the asymptotically justified default in high dimensions.

This distinction is also important for interpreting the practical implications of the mechanism-level comparison. When the additive vector release itself is the private primitive, the MSE reduction in Figure 2 directly corresponds to improved accuracy in the stated parameter regimes. Examples include low-dimensional mean estimation and low-dimensional vector releases under composition. By contrast, when the additive noise mechanism is used as part in a larger algorithmic design, the same mechanism-level gain need not automatically imply an end-to-end improvement. For high-dimensional private learning methods such as DP-SGD, additional work is needed to determine whether any low-dimensional SGG gains translate into improved training utility

### 4.4. Privacy Accounting of SGG Mechanisms

Since additive-noise primitives are often invoked repeatedly in DP computations, we study tight composed privacy for multi-call SGG mechanisms. The key observation is that the privacy loss random variables (PRVs) add under independent composition. Thus, once the single-step PRV distribution of an SGG mechanism can be evaluated, the composed privacy profile can be computed by convolution.

**Proposition 4.2** (Composition accounting for SGG mech-

anisms)**.** *Consider $k$ independent invocations of SGG additive mechanisms, each calibrated for an $\ell_2$-sensitivity bound. Let $Z_1, \ldots, Z_k$ denote their worst-case one-step PRVs, as characterized in Lemma D.1. Then the composed privacy profile at level $\varepsilon_{\text{tot}}$ is*

$$\delta_k(\varepsilon_{\text{tot}}) = \mathbf{Ex}\left[\left(1 - \exp\left(\varepsilon_{\text{tot}} - \sum_{i=1}^{k} Z_i\right)\right)_+\right].$$

*Consequently, any upper bound $\widehat{\delta}_k(\varepsilon_{\text{tot}}) \geq \delta_k(\varepsilon_{\text{tot}})$ provides that the $k$-fold composition is $(\varepsilon_{\text{tot}}, \widehat{\delta}_k(\varepsilon_{\text{tot}}))$-DP.*

The proof and the SGG-specific reduction of the PRV CDF to a one-dimensional radial integral are given in Appendix D. Algorithm 7 implements the resulting accountant by discretizing the single-step PRV distribution and computing the $k$-fold convolution by FFT. Since the $\ell_2$ mechanism is a special case of SGG, the same accounting framework also gives a tight accountant for composed $\ell_2$ mechanisms.

## 5. Conclusion

We show that as $T \to \infty$, the standard Gaussian mechanism is asymptotically best among all additive noise mechanisms. Also, we provide a new family of SGG mechanisms and specific choices within this family that outperform both the Gaussian and recent $\ell_2$ mechanism in certain low-dimensional settings. These low-dimensional improvements should be viewed as an existence result: they demonstrate nontrivial finite-dimensional design space, but do not show that non-Gaussian SGG consistently provide meaningful improvement over all parameters, dimensions, or downstream applications. Furthermore, we show tight composition of the SGG mechanisms under the $(\varepsilon, \delta)$-differential privacy, and answering an open question of (Joseph et al., 2025) regarding the $\ell_2$ mechanism. We view extending the studies beyond additive noise — for example, to data-dependent noise and to non-additive mechanisms — as an important direction for future work.

## Acknowledgements

We thank the reviewers for their careful reading and insightful comments, which helped us clarify the scope and presentation of the paper. We are also grateful to Hubert Chan for several helpful discussions on the direction and formulation of our main asymptotic optimality result in Theorem 3.1.

## Disclaimer

This paper was prepared for informational purposes in part by the Artificial Intelligence Research group of JPMorgan

## Impact Statement

Our work studies theoretical and algorithmic tools for analyzing and calibrating additive-noise mechanisms under $(\varepsilon, \delta)$-differential privacy. The primary intended impact is to enable practitioners to obtain *tighter* privacy accounting and, consequently, to add less noise for a given privacy target or to certify stronger privacy for a fixed utility level.

We do not anticipate negative societal impacts beyond the general risk that differential privacy may be misapplied in a manner that is inappropriate to the application, threat model, and sensitivity of the data. Additionally, we emphasize that our methods do not prescribe a universal "safe" choice of $(\varepsilon, \delta)$; users should select privacy parameters in consultation with domain requirements and established guidance, and should report these choices transparently.

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

## A. Additional Preliminaries

Proposition A.1 characterizes the distribution of $\cos \Theta$, where $\Theta$ is the angle between the random direction vector $U$ and a fixed reference direction. This random variable appears repeatedly in our analysis. We write $\mathsf{Unif}(\mathcal{S}^{T-1})$ for the uniform distribution on the $T$-dimensional unit sphere.

**Proposition A.1.** *Let random variable $\Theta$ be the angle between a fixed $T$-dimensional vector and a $T$-dimensional random vector $U \sim \mathsf{Unif}(\mathcal{S}^{T-1})$. The density function $f_{\cos \Theta}$ for the random variable $\cos \Theta$ is with the form*

$$f_{\cos \Theta}(u) = \frac{\Gamma(\frac{T}{2})}{\sqrt{\pi}\Gamma(\frac{T-1}{2})}(1-u^2)^{\frac{T-3}{2}}, \quad u \in [-1,1],$$

*and its CDF is with the form [4]*

$$\Pr[\cos \Theta \leq u] = I_{\frac{u+1}{2}}(\frac{T-1}{2}, \frac{T-1}{2}).$$

*Proof.* Let random variable $X = \frac{\cos \Theta + 1}{2} \in [0,1]$ so that $\cos \Theta = 2X - 1$. Applying the change of variable formula for pdf, we have

$$\Pr[X = x] = 2\Pr[\cos \Theta = 2x - 1] = \frac{\Gamma(\frac{T}{2})2^{T-2}}{\sqrt{\pi}\Gamma(\frac{T-1}{2})}(x(1-x))^{\frac{T-3}{2}}.$$

Using Legendre's duplication formula, we observe that

$$\Gamma(\frac{T-1}{2})\Gamma(\frac{T}{2}) = 2^{2-T}\sqrt{\pi}\Gamma(T-1), \quad \Gamma(\frac{T}{2}) = \frac{2^{2-T}\sqrt{\pi}\Gamma(T-1)}{\Gamma(\frac{T-1}{2})}.$$

Plugging it in the expression of the PDF of $X$, we have

$$\begin{aligned}
\Pr[X = x] &= \frac{\frac{2^{2-T}\sqrt{\pi}\Gamma(T-1)}{\Gamma(\frac{T-1}{2})}2^{T-2}}{\sqrt{\pi}\Gamma(\frac{T-1}{2})}x^{\frac{T-3}{2}}(1-x)^{\frac{T-3}{2}} \\
&= \frac{\Gamma(T-1)}{\Gamma(\frac{T-1}{2})^2}x^{\frac{T-3}{2}}(1-x)^{\frac{T-3}{2}} \\
&= \frac{1}{B(\frac{T-1}{2}, \frac{T-1}{2})}x^{\frac{T-1}{2}-1}(1-x)^{\frac{T-1}{2}-1},
\end{aligned}$$

which is the density function of beta distribution with parameter $\alpha = \frac{T-1}{2}, \beta = \frac{T-1}{2}$.

Finally, using the CDF of the beta distribution, we have

$$\Pr[\cos \Theta \leq u] = \Pr[2X - 1 \leq u] = \Pr\left[X \leq \frac{u+1}{2}\right] = I_{\frac{u+1}{2}}(\frac{T-1}{2}, \frac{T-1}{2}).$$

$\square$

## B. Proof of Theorem 3.1

*Proof of Lemma 3.1.* Recall the definition of optimal $\delta$ ( Equation (2)) between $X, Y$ at privacy level $\varepsilon$ as

$$\delta = \sup_{\mathcal{S} \subseteq \mathbb{R}^T} \left(\Pr[X \in \mathcal{S}] - e^{\varepsilon}\Pr[Y \in \mathcal{S}]\right)_+.$$

We will explicitly find a set $\mathcal{S}_T$ such that $\left(\Pr[X \in \mathcal{S}] - e^{\varepsilon}\Pr[Y \in \mathcal{S}]\right)_+$ approaches $\mathbf{Ex}\left[g\left(\frac{R_T^2}{T}\right)\right]$.

---

[4]$I_x(a,b) = \frac{\Gamma(a+b)}{\Gamma(a)\Gamma(b)}\int_0^x t^{a-1}(1-t)^{b-1}\mathrm{d}t$ is the regularised incomplete beta function.

By rotational invariance of $U_T \sim \mathsf{Unif}(\mathcal{S}^{T-1})$, WLOG, set $\mu_T = se_1$ (where $e_1 = (1, 0, \ldots, 0)$). Then, we set

$$X_T = R_T U_T, \qquad Y_T = R_T U_T + se_1.$$

We then define the set $\mathcal{S}_T$ as the following

$$\mathcal{S}_T = \left\{ x \in \mathbb{R}^T : \langle e_1, x \rangle \le \frac{s}{2} - \frac{\varepsilon \|x\|_2^2}{sT} \right\}.$$

By definition of $\delta$, we know that $\delta \ge \Pr[X \in \mathcal{S}_T] - e^\varepsilon \Pr[Y \in \mathcal{S}_T]$.

In the rest of the proof, we will study the asymptotics of the expression $\Pr[X \in \mathcal{S}_T] - e^\varepsilon \Pr[Y \in \mathcal{S}_T]$. First, look at the asymptotics of the first coordinate of a random direction $\langle e_1, U_T \rangle$. Define

$$W_T = \sqrt{T} \langle e_1, U_T \rangle.$$

Using the Gaussian representation $U_T \overset{d}{=} G/\|G\|_2$ with $G \sim \mathcal{N}(0, I_T)$, we have that $W_T = \frac{G_1}{\|G\|_2 / \sqrt{T}}$ converges in distribution to a standard normal distribution $\mathcal{N}(0, 1)$. By Pólya's theorem, we have that

$$\lim_{T \to \infty} \sup_{t \in \mathbb{R}} |\Pr[W_T \le t] - \Phi(t)| = 0, \tag{9}$$

where $\Phi$ is the standard normal cdf.

Then, we study the asymptotics of the term $\Pr[X_T \in \mathcal{S}_T \mid R_T = r]$. Condition on $R_T = r$, we observe that $X_T = rU_T$ and $\|X\|_2^2 = r^2$ is deterministic given $r$. Therefore, by the definition of $\mathcal{S}_T$,

$$X_T \in \mathcal{S}_T \iff r \langle e_1, U \rangle \le \frac{s}{2} - \frac{\varepsilon}{s} \cdot \frac{r^2}{T} \iff W_T \le a\left(\frac{r^2}{T}\right),$$

where we denote function $a(u)$ for $u > 0$:

$$a(u) = -\frac{\varepsilon \sqrt{u}}{s} + \frac{s}{2\sqrt{u}}.$$

Using Equation (9), we have that

$$\Pr[X_T \in \mathcal{S}_T \mid R_T = r] = \Phi\left(a\left(\frac{r^2}{T}\right)\right) + o(1),$$

where the $o(1)$ is uniform in $r$.

Similarly, look at the asymptotics of the term $\Pr[Y_T \in \mathcal{S}_T \mid R_T = r]$. Recall $Y_T = rU_T + se_1$ and $\langle e_1, Y_T \rangle = rU_{T,1} + s$. Then,

$$\begin{aligned}
\frac{\|Y\|_2^2}{T} &= \frac{r^2 + s^2 + 2rs\langle e_1, U_T \rangle}{T} \\
&= \frac{r^2}{T} + \frac{s^2}{T} + \frac{2rs}{T} \cdot \frac{W_T}{\sqrt{T}} \\
&= \frac{r^2}{T} + O_p\left(\frac{1}{T}\right) \\
&= \frac{r^2}{T} + o_p(1),
\end{aligned}$$

where $O_p, o_p$ is the standard notation for asymptotic stochastic boundedness. Therefore, by the definition of $\mathcal{S}_T$,

$$Y_T \in \mathcal{S}_T \iff r\langle e_1, U\rangle \leq -\frac{s}{2} - \frac{\varepsilon}{s} \cdot \frac{r^2}{T} + O_p\left(\frac{1}{T}\right)$$

$$\iff W_T \leq b\left(\frac{r^2}{T}\right) + O_p\left(\frac{1}{\sqrt{T}}\right)$$

$$\iff W_T \leq b\left(\frac{r^2}{T}\right) + o_p(1),$$

where we denote function $b(u)$ for $u > 0$:

$$b(u) = -\frac{\varepsilon\sqrt{u}}{s} - \frac{s}{2\sqrt{u}}.$$

Using Equation (9), we have that

$$\Pr\left[Y_T \in \mathcal{S}_T \mid R_T = r\right] = \Phi\left(b\left(\frac{r^2}{T}\right)\right) + o(1),$$

where the $o(1)$ is uniform in $r$.

Taking expectation of $\Pr\left[Y_T \in \mathcal{S}_T \mid R_T = r\right]$ and $\Pr\left[X_T \in \mathcal{S}_T \mid R_T = r\right]$ over $R_T$ and plugging into $\Pr\left[X \in \mathcal{S}_T\right] - e^\varepsilon \Pr\left[Y \in \mathcal{S}_T\right]$ gives

$$\Pr\left[X \in \mathcal{S}_T\right] - e^\varepsilon \Pr\left[Y \in \mathcal{S}_T\right]$$

$$= \mathbf{Ex}\left[\Phi\left(a\left(\frac{R_T^2}{T}\right)\right) - e^\varepsilon \Phi\left(b\left(\frac{R_T^2}{T}\right)\right)\right] + o(1),$$

where the expression $\Phi\left(a\left(\frac{R_T^2}{T}\right)\right) - e^\varepsilon \Phi\left(b\left(\frac{R_T^2}{T}\right)\right)$ is exactly the optimal delta of the Gaussian mechanism $\mathcal{M}_{\mathsf{G},u}$ at privacy level $\varepsilon$ (cf. Equation (6)).

Finally, recall that $\delta \geq \left(\Pr\left[X \in \mathcal{S}_T\right] - e^\varepsilon \Pr\left[Y \in \mathcal{S}_T\right]\right)_+$, we finish the proof. $\qquad \square$

*Proof of Proposition 3.1.* By convexity of $g$ on $[u_0, \infty)$, the tangent line at $u_0$ is a global under-estimator on the right, namely, for all $u \geq u_0$,

$$g(u) \geq g(u_0) + g'(u_0)(u - u_0) = \ell_\delta(u),$$

Since we assume that $g(u) \geq \ell_\delta(u)$ for all $u \in [0, u_0]$ in the statement, we have $g(u) \geq \ell_\delta(u)$ for all $u \geq 0$.

Taking expectation with respect to any $U \geq 0$ yields

$$\mathbf{Ex}\left[g(U)\right] \geq \mathbf{Ex}\left[\ell_\delta(U)\right]$$
$$= \mathbf{Ex}\left[g(u_0) + g'(u_0)(U - u_0)\right]$$
$$= g(u_0) + g'(u_0)(\mathbf{Ex}\left[U\right] - u_0)$$
$$= g(u_0).$$

$\qquad \square$

**Proposition B.1** (Downward closure of the tangent-support conditions). *Fix $\varepsilon \geq 0$, $\delta \in (0,1)$, $s > 0$, and let $g : (0, \infty) \to \mathbb{R}$ be differentiable and strictly decreasing. For any $\delta \in (0,1)$, Let $u(\delta) > 0$ be the unique point such that $g(u(\delta)) = \delta$ and define the tangent line function at $t > 0$ by*

$$\ell_t(u) = g(t) + g'(t)(u - t).$$

*If there exists a $\delta_0 \in (0,1)$ and $u_0 = u(\delta_0)$ such that: $g$ is convex on $[u_0, \infty)$; and $g(u) \geq \ell_{u_0}(u)$ for all $u \in [0, u_0]$. Then the two conditions hold is downward closed for all $\delta \in (0, \delta_0)$.*

*Proof.* Fix any $\delta \in (0, \delta_0)$ and let $u_1 = u(\delta)$. Since $g$ is strictly decreasing and $g(u_0) = \delta_0 > \delta = g(u_1)$, it follows that $u_1 > u_0$.

It is easy to see that the first condition holds on $u_1$. Because $[u_1, \infty) \subseteq [u_0, \infty)$ and $g$ is convex on $[u_0, \infty)$, it is convex on the subset $[u_1, \infty)$ as well.

To show the second condition holds, we split the interval $[0, u_1]$ into two parts and consider each case separately. For any fixed $u \in [0, u_0]$, define function $F(t) = l_t(u)$ for $t \geq u$, whose output is the tangent line function at $t$ evaluating at a fixed point $u$. Differentiating $F(t)$ with respect to $t$, we have

$$F'(t) = g'(t) + g''(t)(u - t) - g'(t) = g''(t)(u - t).$$

On $[u_0, \infty)$ we have $g''(t) \geq 0$ by convexity, and since $t \geq u$ we have $u - t \leq 0$. Hence $F'(t) \leq 0$ for all $t \geq u_0$, so $F$ is nonincreasing on $[u_0, \infty)$. Because $u_1 \geq u_0$, $\ell_{u_1}(u) = F(u_1) \leq F(u_0) = \ell_{u_0}(u)$. By assumption, $g(u) \geq \ell_{u_0}(u)$ for all $u \in [0, u_0]$, thus

$$g(u) \geq \ell_{u_0}(u) \geq \ell_{u_1}(u), \qquad \forall u \in [0, u_0].$$

For the second case, where $u \in [u_0, u_1]$, $h(u) = g(u) - \ell_{u_1}(u)$. We know that $h'(u) = g'(u) - g'(u_1)$. Since $g$ is decreasing and convex on $[u_0, \infty)$, its derivative $g'$ is nondecreasing there. For any $u_0 \leq u \leq u_1$, this implies $g'(u) \leq g'(u_1)$, hence $h'(u) \leq 0$ on $[u_0, u_1]$. Therefore $h$ is nonincreasing on $[u_0, u_1]$, and for all $u \in [u_0, u_1]$ we have

$$h(u) \geq h(u_1) = 0,$$

i.e., $g(u) \geq \ell_{u_1}(u)$.

Combining both cases yields $g(u) \geq \ell_{u_1}(u)$ for all $u \in [0, u_1]$, completing the proof for the second condition. $\qquad\square$

*Proof is in Lemma 3.2.* Denote function $\phi(x) = \frac{1}{\sqrt{2\pi}} e^{-x^2/2}$ as the standard normal density. We start from the case where $\varepsilon > 0$. Our first step is to give an asymptotic form for function $g(u)$, in which the asymptotics is in terms of $u$. Concretely,

$$g(u) = C_{\varepsilon,s} u^{-3/2} \exp\left(-\frac{\varepsilon^2}{2s^2} u\right) \left(1 + O\left(\frac{1}{u}\right)\right), \tag{10}$$

with $C_{\varepsilon,s} = \frac{e^{\varepsilon/2} s^3}{\sqrt{2\pi}\, \varepsilon^2}$.

Let $x = x(u) = \frac{\varepsilon\sqrt{u}}{s}$, and $d = d(u) = \frac{s}{2\sqrt{u}} = \frac{\varepsilon}{2x}$. We rewrite $g(u)$ as $g(u) = \Phi\big(-(x - d)\big) - e^\varepsilon \Phi\big(-(x + d)\big)$. The standard Mills bounds says that for all $z > 0$,

$$\frac{1}{z + z^{-1}} \leq \frac{\Phi(-z)}{\phi(z)} \leq \frac{1}{z}.$$

Then we have

$$0 \leq \frac{1}{z} - \frac{\Phi(-z)}{\phi(z)} \leq \frac{1}{z} - \frac{1}{z + z^{-1}} = \frac{1}{z^3 + z} \leq \frac{1}{z^3},$$

which gives us $\frac{\Phi(-z)}{\phi(z)} = \frac{1}{z} + O(z^{-3})$ as $z \to \infty$.

Define $F(z) = \frac{\Phi(-z)}{\phi(z)}$. Using $\Phi(-z) = \phi(z) F(z)$ and the identity $\phi(x - d) = e^\varepsilon \phi(x + d)$, we express $g(u)$ as

$$g(u) = \phi(x - d)\big(F(x - d) - F(x + d)\big).$$

Applying the asymptotic form of $F(z)$ at $z = x \pm d$, and use that $d = \varepsilon/(2x) = O(1/x)$ as $x \to \infty$:

$$\begin{aligned}
F(x - d) - F(x + d) &= \left(\frac{1}{x - d} - \frac{1}{x + d}\right) + O\left(\frac{1}{x^3} \cdot \frac{d}{x}\right) \\
&= \frac{2d}{x^2 - d^2} + O\left(\frac{1}{x^5}\right) \\
&= \frac{\varepsilon}{x^3} + O\left(\frac{1}{x^5}\right).
\end{aligned}$$

Moreover,

$$
\begin{aligned}
\phi(x - d) &= \frac{1}{\sqrt{2\pi}} \exp\left(-\frac{(x-d)^2}{2}\right) \\
&= \frac{1}{\sqrt{2\pi}} \exp\left(-\frac{x^2}{2} + \frac{\varepsilon}{2} - \frac{d^2}{2}\right) \\
&= \frac{e^{\varepsilon/2}}{\sqrt{2\pi}} e^{-x^2/2}\left(1 + O\left(\frac{1}{x^2}\right)\right).
\end{aligned}
$$

Plugging these into $g(u) = \phi(x - d)\big(F(x - d) - F(x + d)\big)$ yields

$$
g(u) = \frac{e^{\varepsilon/2}}{\sqrt{2\pi}} e^{-x^2/2}\left(\frac{\varepsilon}{x^3} + O\left(\frac{1}{x^5}\right)\right).
$$

Finally, substitute $x$ and $d$ in terms of $\varepsilon, u, s$ back, we obtain the asymptotic form of $g(u)$ as expressed in Equation (10).

We then show that for the case $\varepsilon > 0$, there exists $u_{\text{right}} > 0$ such that $g$ is convex on $[u_{\text{right}}, \infty)$. Let $k = \varepsilon^2/(2s^2) > 0$, we first rewrite $g(u) = f(u)(1 + \eta(u))$, where $f(u) = C_{\varepsilon,s} u^{-3/2} e^{-ku}$ and $\eta(u) = O(1/u)$ by Equation (10).

Differentiating $(\log g)(u) = (\log f)(u) + \log(1 + \eta(u))$ with respect to $u$, we have

$$
\begin{aligned}
(\log g)'(u) &= -k - \frac{3}{2u} + O\left(\frac{1}{u^2}\right) \\
(\log g)''(u) &= \frac{3}{2u^2} + O\left(\frac{1}{u^3}\right).
\end{aligned}
\tag{11}
$$

Using the identity $g(u) = \exp\left(\log g(u)\right)$, and differentiate both side with respect to $u$ twice, we have

$$
g''(u) = g(u)\Big((\log g)'(u)^2 + (\log g)''(u)\Big).
$$

Plugging Equation (11) into the expression of $g''(u)$ above, we have

$$
g''(u) = g(u)\left(k^2 + O\left(\frac{1}{u}\right)\right).
$$

Since $g(u) > 0$ for all $u > 0$, it is clear to see that there exists $u_{\text{right}} > 0$ such that $g''(u) > 0$ for all $u \geq u_{\text{right}}$, i.e. $g$ is convex on $[u_{\text{right}}, \infty)$.

Next, we define the function $L(u)$ as the y-intercept of the tangent line at $u$ (recall the tangent line is defined in the statement $\ell_\delta(u)$, and $u$ is the $u_0(\delta)$ as defined). Concretely,

$$
L(u) = g(u) - ug'(u) = g(u)\big(1 - u(\log g)'(u)\big).
$$

Plugging Equation (11) into the above, we have $L(u) = g(u)(1 + ku + O(1))$. Since $g(u)$ decays like $u^{-3/2}e^{-ku}$, we have

$$
\lim_{u \to \infty} L(u) = 0.
$$

We now are ready to show that for the case $\varepsilon > 0$, there exists a $u_\star$ that satisfies both conditions. Let $m = g(u_{\text{right}})$, and since $g$ is a decreasing function, we know that $m \leq g(u)$ for all $u \in [0, u_{\text{right}}]$. Choose $u_\star \geq u_{\text{right}}$ large enough that

$$
L(u_\star) \leq m.
$$

Let $\delta_\star = g(u_\star) \in (0, 1)$ and define the intercept function at $u_\star$ as $\ell_\star(u) = g(u_\star) + g'(u_\star)(u - u_\star)$.

We claim that $g(u) \geq \ell_\star(u)$ for all $u \in [0, u_\star]$. First, for $u \in [0, u_{\text{right}}]$, since function $\ell_\star(u)$ is decreasing in $u$, we have that

$$\ell_\star(u) \leq \ell_\star(0) = g(u_\star) - u_\star g'(u_\star) \leq m \leq g(u).$$

Second, for $u \in [u_{\text{right}}, u_\star]$, convexity of $g$ on $[u_{\text{right}}, \infty)$ implies that $g(u) \geq \ell_\star(u)$. Therefore we prove that for the case $\varepsilon > 0$, there exists a point $u_\star$ such that $g(u) \geq \ell_\star(u)$ on $[0, u_\star]$, and $g$ is convex on $[u_\star, \infty)$.

We notice that the existence also holds for the case $\varepsilon = 0$. Using Taylor expansion of $\Phi$ at 0, we can derive an asymptotic form for function $g(u)$ when $\varepsilon = 0$:

$$g(u) = 2\left(\frac{1}{2} + \phi(0)t + O(t^3)\right) - 1 = \frac{s}{\sqrt{2\pi}}u^{-1/2} + O(u^{-3/2}).$$

Also, we have $L(u) = g(u) - ug'(u) = \frac{3s}{2\sqrt{2\pi}}u^{-1/2} + o(u^{-1/2}) \to 0$ as $u \to \infty$. The rest will follow directly from the proof for the case $\varepsilon > 0$.

To finish the proof, we then apply the downward-closure argument ( Proposition B.1), and observe that the two conditions that hold at $u_\star$ also hold at $u_0 \geq u_\star$: $g$ is convex on $[u_0, \infty)$ and $g(u) \geq \ell_\delta(u)$ for all $u \in [0, u_0]$.

This proves the lemma. □

Combining Lemmas 3.1 and 3.2 and proposition 3.1 yields the conclusion that, in the high-privacy regime, the Gaussian mechanism is asymptotically optimal among all *spherical* additive-noise mechanisms with the same MSE. Theorem B.1 formlizes this.

**Theorem B.1** (Asymptotic Gaussian optimality among spherical noises). *Fix $\varepsilon \geq 0$ and $s > 0$. There exists $\delta_\star \in (0, 1)$ such that for every $\delta \in (0, \delta_\star)$ the following holds. Let $u_0 = u_0(\delta)$ be the minimal Gaussian variance for which the Gaussian mechanism $\mathcal{M}_{G, u_0}$ is $(\varepsilon, \delta)$-DP. For each $T \geq 2$, let $\mathcal{M}_{R_T, T, u_0}$ be any spherically symmetric additive-noise mechanism with*

$$\mathbf{Ex}\left[R_T^2\right] = Tu_0.$$

*Then*

$$\liminf_{T \to \infty} \delta_{\mathcal{M}_{R_T, T, u_0}}(\varepsilon) \geq \delta_{\mathcal{M}_{G, u_0}}(\varepsilon) = \delta.$$

*Equivalently, for every $\eta > 0$, for all sufficiently large $T$,*

$$\delta_{\mathcal{M}_{R_T, T, u_0}}(\varepsilon) \geq \delta - \eta.$$

*Proof.* Fix $\varepsilon \geq 0$, $s > 0$, and let $\delta_\star$ be as in Lemma 3.2. Fix any $\delta \in (0, \delta_\star]$, and let $u_0 = u_0(\delta) > 0$ be the (unique) solution to $g(u_0) = \delta$. Recall that, we define Gaussian privacy function $g$ for $u > 0$ (cf. Equation (6)),

$$g(u) = \Phi\left(-\frac{\varepsilon\sqrt{u}}{s} + \frac{s}{2\sqrt{u}}\right) - e^\varepsilon \Phi\left(-\frac{\varepsilon\sqrt{u}}{s} - \frac{s}{2\sqrt{u}}\right).$$

Now consider the spherical mechanism $\mathcal{M}_{R_T, T, u_0}(\mu_T) = \mu_T + R_T U_T$ with $\mathbf{Ex}\left[R_T^2\right] = Tu_0$. Let $\mu_T \in \mathbb{R}^T$ be any shift with $\|\mu_T\|_2 = s$, and define neighboring output distribution

$$X_T = R_T U_T, \qquad Y_T = X_T + \mu_T.$$

Let $\delta_T^{\text{hid}}(\varepsilon)$ denote the optimal delta at level $\varepsilon$ for the pair $(X_T, Y_T)$. By definition of $\delta_\varepsilon(\mathcal{M}_{R_T, T, u_0})$ as the worst-case delta over neighboring databases, we know that for every choice of $\mu_T$ with $\|\mu_T\|_2 = s$, we have that

$$\delta_\varepsilon(\mathcal{M}_{R_T, T, u_0}) \geq \delta_T^{\text{hid}}(\varepsilon).$$

We compare the privacy parameter $\delta_T^{\text{hid}}(\varepsilon)$ to that of the Gaussian benchmark $\mathcal{M}_G$.

Let $U = R_T^2/T$. By Lemma 3.1, we have

$$\delta_T^{\text{hid}}(\varepsilon) \geq \mathbf{Ex}\left[g(U)\right] + o(1)$$
$$= \mathbf{Ex}\left[g\left(\frac{R_T^2}{T}\right)\right] + o(1), \qquad T \to \infty,$$

where $o(1) \to 0$ as $T \to \infty$ for fixed $(\varepsilon, s)$.

It remains to lower bound $\mathbf{Ex}\left[g\left(\frac{R_T^2}{T}\right)\right]$ under the second-moment constraint $\mathbf{Ex}\left[\frac{R_T^2}{T}\right] = u_0$. Apply Proposition 3.1 with the random variable $U$. By Lemma 3.2 (with $\delta \leq \delta_\star$), the function $g$ satisfies the convexity and tangent-line conditions at $u_0$, hence Proposition 3.1 yields

$$\mathbf{Ex}\left[g(U)\right] \geq g(u_0) = \delta. \tag{12}$$

Combining the above two inequalities gives

$$\delta_T^{\text{hid}}(\varepsilon) \geq \mathbf{Ex}\left[g(U)\right] + o(1) \geq g(u_0) + o(1) = \delta + o(1).$$

Therefore,

$$\liminf_{T\to\infty} \delta_\varepsilon(\mathcal{M}_{R_T,T,u_0}) \geq \liminf_{T\to\infty} \delta_T^{\text{hid}}(\varepsilon) \geq \delta,$$

which is exactly the claimed statement. $\square$

*Proof of Lemma 3.3.* Recall that, by Lemma 3.3's definition, $M$ is a Haar-uniform random matrix over the group of all $T \times T$ orthogonal matrices and independent of a random vector $X \in \mathbb{R}^T$. Let $X' = MX$.

We first show that $X'$ is spherically symmetric. Fix any deterministic $T \times T$ orthogonal matrix $Q$. We know that $QM \overset{d}{=} M$ by Haar-uniform random matrix's property. Therefore, we have

$$QX' = QMX \overset{d}{=} MX = X',$$

which, by definition, says that $X'$ is spherically symmetric.

Since orthogonal transformations preserve the Euclidean norm, so $\|X'\|_2 = \|MX\|_2 = \|X\|_2$, and therefore

$$\mathbf{Ex}\left[\|X'\|_2^2\right] = \mathbf{Ex}\left[\|X\|_2^2\right].$$

It remains to prove $\delta' \leq \delta$. Fix any $\mu \in \mathbb{R}^T$ with $\|\mu\|_2 \leq s$. Let $\nu$ denote Haar measure over the orthogonal group. For each orthogonal matrix $O$, define $P_O, Q_O$ as the distribution of $OX$ and $OX + \mu$, respectively. By convexity of hockey-stick divergence under mixtures,

$$\mathsf{H}_\varepsilon\left(X', X' + \mu\right) \leq \int \mathsf{H}_\varepsilon\left(P_O, Q_O\right) d\nu(O).$$

For every fixed orthogonal matrix $O$, applying the bijection $x \mapsto O^\intercal x$ to both distributions gives

$$\mathsf{H}_\varepsilon\left(OX, OX + \mu\right) = \mathsf{H}_\varepsilon\left(X, X + O^\intercal\mu\right).$$

Since $\|O^\intercal\mu\|_2 = \|\mu\|_2 \leq s$, the definition of $\delta$ gives $\mathsf{H}_\varepsilon\left(X, X + O^\intercal\mu\right) \leq \delta$. Therefore,

$$\mathsf{H}_\varepsilon\left(X', X' + \mu\right) \leq \delta.$$

Taking the supremum over all $\mu$ with $\|\mu\|_2 \leq s$ gives $\delta' \leq \delta$. $\square$

To finish the proof of Theorem 3.1, we further use Lemma 3.3, which formalize that, under a fixed MSE budget, the optimal privacy achievable by additive-noise mechanisms is no better than that achievable by spherical additive-noise mechanisms.

*Proof of Theorem 3.1.* Fix $\varepsilon \geq 0$ and $s > 0$, and let $\delta_\star$ be as in Lemma 3.2. Fix any $\delta \in (0, \delta_\star]$, and let $u_0 = u_0(\delta) > 0$ be the unique solution to $g(u_0) = \delta$, where (cf. Equation (6))

$$g(u) = \Phi\left(-\frac{\varepsilon\sqrt{u}}{s} + \frac{s}{2\sqrt{u}}\right) - e^\varepsilon \Phi\left(-\frac{\varepsilon\sqrt{u}}{s} - \frac{s}{2\sqrt{u}}\right).$$

Now fix any $T \geq 2$ and consider an arbitrary additive-noise mechanism

$$\mathcal{M}_{T,u_0}(\mu_T) = \mu_T + X_T,$$

where $X_T \in \mathbb{R}^T$ is a random vector satisfying the MSE constraint $\mathbf{Ex}\left[\|X_T\|_2^2\right] = Tu_0$. Let

$$\delta_T(\varepsilon) \overset{\text{def}}{=} \sup_{\|v\|_2 \leq s} \mathsf{H}_\varepsilon\left(X_T, X_T + v\right).$$

By definition of the worst-case optimal delta for the additive-noise mechanism, $\delta_{\mathcal{M}_{T,u_0}}(\varepsilon) = \delta_T(\varepsilon)$. Let $X'_T = M_T X_T$ be the symmetrized noise of $X_T$, where $M_T$ is the Haar-uniform random matrix over the group of all $T \times T$ orthogonal matrices. Define

$$\delta'_T(\varepsilon) \overset{\text{def}}{=} \sup_{\|v\|_2 \leq s} \mathsf{H}_\varepsilon\left(X'_T, X'_T + v\right).$$

By Lemma 3.3, we know $X'_T$ is spherically symmetric, has the same MSE as $X_T$, and moreover

$$\delta'_T(\varepsilon) \leq \delta_T(\varepsilon). \tag{13}$$

Since $X'_T$ is spherically symmetric, there exists a nonnegative random variable $R_T > 0$ and an independent $U_T \sim \mathsf{Unif}(\mathcal{S}^{T-1})$ such that $X'_T \overset{d}{=} R_T U_T$, and $\mathbf{Ex}\left[R_T^2\right] = Tu_0$. Therefore, the spherical optimality result Theorem B.1 applies to the spherical mechanism $\mu_T + X'_T$ and yields

$$\liminf_{T\to\infty} \delta'_T(\varepsilon) \geq \delta_{\mathcal{M}_{\mathsf{G},u_0}}(\varepsilon) = \delta. \tag{14}$$

Combining Equation (13) and Equation (14) gives

$$\liminf_{T\to\infty} \delta_{\mathcal{M}_{T,u_0}}(\varepsilon) = \liminf_{T\to\infty} \delta_T(\varepsilon) \geq \liminf_{T\to\infty} \delta'_T(\varepsilon) \geq \delta.$$

This is exactly the claimed statement. $\qquad\square$

## C. Supplementary Material for Spherical Generalized Gamma Mechanism

### C.1. Numerical Bound

*Proof of Lemma 4.1.* Let $Y_0, Y_1, X$ be i.i.d. with $X = RU \sim \mathsf{SGG}(\alpha, \beta, p)$, where $R \sim \mathsf{GGamma}(\alpha, \beta, p)$ and $U \sim \mathsf{Unif}(\mathcal{S}^{T-1})$ are independent. Define $\Theta$ the angle between $\mu$ and $U$, and recall $W = \cos\Theta \in [-1, 1]$, whose CDF $F_W$ given in Proposition A.1.

Let $f_X$ denote the density of $X$. Since $Y_0 \overset{d}{=} X$ and $Y_1 + \mu \overset{d}{=} X + \mu$, we have $f_{Y_0}(x) = f_X(x)$ and $f_{Y_1+\mu}(x) = f_X(x-\mu)$. By the privacy-loss random variable characterization of the optimal $\delta(\varepsilon)$ by a fixed $\varepsilon$ ( (Balle & Wang, 2018a)),

$$\delta^*(\varepsilon) = \left[\Pr\left(L^{\mathsf{neg}}(X) \leq -\varepsilon\right) - e^\varepsilon \Pr\left(L^{\mathsf{pos}}(X) \geq \varepsilon\right)\right]_+,$$

where

$$L^{\mathsf{pos}}(X) = \ln\frac{f_{Y_1+\mu}(X)}{f_{Y_0}(X)} = \ln\frac{f_X(X-\mu)}{f_X(X)},$$

$$L^{\mathsf{neg}}(X) = \ln\frac{f_{Y_0}(X+\mu)}{f_{Y_1+\mu}(X+\mu)} = \ln\frac{f_X(X+\mu)}{f_X(X)}.$$

**Spherical symmetry reduces the PLRV to a function of $(R, W)$.** By Theorem 2.1, the SGG density is spherically symmetric, i.e., $f_X(x) = g(\|x\|_2^2)$ for a suitable generator $g$. Therefore,

$$L^{\mathsf{pos}}(X) = \ln \frac{g(\|X - \mu\|_2^2)}{g(\|X\|_2^2)} = \ln \frac{g(R^2 - 2RsW + s^2)}{g(R^2)},$$

$$L^{\mathsf{neg}}(X) = \ln \frac{g(\|X + \mu\|_2^2)}{g(\|X\|_2^2)} = \ln \frac{g(R^2 + 2RsW + s^2)}{g(R^2)}.$$

Moreover, since $U \sim \mathsf{Unif}(\mathcal{S}^{T-1})$ implies $W \overset{d}{=} -W$, we have $L^{\mathsf{pos}}(R, W) = L^{\mathsf{neg}}(R, -W)$ and hence $L^{\mathsf{pos}}(X) \overset{d}{=} L^{\mathsf{neg}}(X)$. Consequently, we may work with the single privacy loss

$$L(X) = L^{\mathsf{neg}}(X),$$

and write

$$\delta^*(\varepsilon) = \Big[\Pr\left[L(X) \leq -\varepsilon\right] - e^\varepsilon \Pr\left[L(X) \geq \varepsilon\right]\Big]_+.$$

For $X \sim \mathsf{SGG}(\alpha, \beta, p)$, Table 1 yields

$$f_X(x) \propto \|x\|_2^{\alpha+1-T} e^{-\beta\|x\|_2^p}, \qquad \text{so} \qquad g(t) \propto t^{\frac{\alpha+1-T}{2}} e^{-\beta t^{p/2}}.$$

Substituting this $g$ into $L(X) = \ln\big(g(\|X + \mu\|_2^2)/g(\|X\|_2^2)\big)$ gives

$$L(X) = \ell(R, W),$$

where for $r > 0$ and $w \in [-1, 1]$,

$$\ell(r, w) = \frac{\alpha + 1 - T}{2} \ln\Big(1 + \frac{2sw}{r} + \frac{s^2}{r^2}\Big) + \beta\Big[r^p - (r^2 + 2swr + s^2)^{p/2}\Big].$$

**Monotonicity in $w$ (for $\alpha \leq T - 1$).** Since the lemma assumes $\alpha \in (-1, T-1]$, we have $\alpha + 1 - T \leq 0$. Let $q(r, w) = r^2 + 2swr + s^2 > 0$. A direct calculation yields

$$\frac{\partial \ell}{\partial w}(r, w) = sr\left[\frac{\alpha + 1 - T}{q(r, w)} - \beta p\, q(r, w)^{\frac{p}{2}-1}\right] = \frac{sr}{q(r, w)}\left[(\alpha + 1 - T) - \beta p\, q(r, w)^{p/2}\right].$$

Because $sr/q(r, w) > 0$, $\beta p\, q(r, w)^{p/2} > 0$, and $(\alpha + 1 - T) \leq 0$, the bracketed term is strictly negative for all $r > 0$ and $w \in [-1, 1]$. Hence $\partial\ell/\partial w(r, w) < 0$ on $(0, \infty) \times [-1, 1]$, so for each fixed $r > 0$ the map $w \mapsto \ell(r, w)$ is strictly decreasing on $[-1, 1]$. Therefore, for any $y \in \mathbb{R}$ and $r > 0$ there is a unique threshold $w^*(r, y) \in [-1, 1]$ satisfying $\ell(r, w^*(r, y)) = y$.

**One-dimensional integral expressions.** Fix $y \in \mathbb{R}$. By strict monotonicity, conditional on $R = r$,

$$\{\ell(r, W) \leq y\} \iff \{W \geq w^*(r, y)\}, \qquad \{\ell(r, W) \geq y\} \iff \{W \leq w^*(r, y)\}.$$

Using independence of $R$ and $W$,

$$\Pr(L(X) \leq -\varepsilon) = \int_0^\infty f_R(r)\big(1 - F_W(w^*(r, -\varepsilon))\big)\, \mathrm{d}r,$$

$$\Pr(L(X) \geq \varepsilon) = \int_0^\infty f_R(r)\, F_W(w^*(r, \varepsilon))\, \mathrm{d}r.$$

Substituting into $\delta^*(\varepsilon) = \big[\Pr(L \leq -\varepsilon) - e^\varepsilon \Pr(L \geq \varepsilon)\big]_+$ yields the desired one-dimensional representation stated in Lemma 4.1. $\qquad\square$

*Proof of Proposition 4.1.* By Theorem 2.1 and Definition 4.1, for $y = \|x\|_2 > 0$ we have

$$f_X(x) = g(y^2) = C_{\alpha,\beta,p,T}\, y^{\alpha+1-T} \exp\!\left(-\beta y^p\right),$$

for a constant $C_{\alpha,\beta,p,T} > 0$ that does not depend on $y$. Differentiate with respect to $y$:

$$\frac{\mathrm{d}}{\mathrm{d}y} g(y^2) = g(y^2)\left(\frac{\alpha+1-T}{y} - \beta p\, y^{p-1}\right).$$

Since $\alpha \leq T-1$ implies $\alpha+1-T \leq 0$, and since $\beta p\, y^{p-1} \geq 0$ for $y > 0$, the bracketed term is non-positive for all $y > 0$. Therefore $\frac{\mathrm{d}}{\mathrm{d}y} g(y^2) \leq 0$ for all $y > 0$, i.e., $y \mapsto g(y^2)$ is non-increasing. This is equivalent to $t \mapsto g(t)$ being non-increasing on $(0,\infty)$ under the change of variables $t = y^2$. $\qquad\square$

*Proof of Lemma 4.2.* Write $P_\mu$ for the distribution of $X + \mu$ and $P_0$ for the distribution of $X$. Since $X$ is spherically symmetric, the quantity $\delta_\mu^*(\varepsilon)$ depends on $\mu$ only through $s = \|\mu\|_2$; hence it suffices to prove monotonicity in $s$. Let $f$ denote the density of $X$ and note that $P_\mu$ has density $f_\mu(x) = f(x - \mu)$. Recall that the optimal $\delta$ at level $\varepsilon$ can be expressed as

$$\delta_\mu^*(\varepsilon) = \int_{\mathbb{R}^T} \big(f(x) - e^\varepsilon f(x-\mu)\big)_+ \,\mathrm{d}x.$$

Equivalently, let $c = e^\varepsilon$ and using $(a-b)_+ = a - \min\{a,b\}$ for $a,b \geq 0$, we may rewrite

$$\delta_\mu^*(\varepsilon) = 1 - \int_{\mathbb{R}^T} \min\{f(x),\, cf(x-\mu)\}\,\mathrm{d}x.$$

Using the identity $\min\{a,b\} = \int_0^\infty \mathbf{1}\{a \geq t\}\mathbf{1}\{b \geq t\}\,\mathrm{d}t$ for $a,b \geq 0$, we have

$$\int_{\mathbb{R}^T} \min\{f(x),\, cf(x-\mu)\}\,\mathrm{d}x = \int_0^\infty \lambda\Big(\{x : f(x) \geq t\} \cap \{x : cf(x-\mu) \geq t\}\Big)\mathrm{d}t$$
$$= \int_0^\infty \lambda\Big(A_t \cap (\mu + A_{t/c})\Big)\mathrm{d}t,$$

where $\lambda(\cdot)$ denotes Lebesgue measure and $A_t = \{x : f(x) \geq t\}$.

By Proposition 4.1, the radial profile $y \mapsto g(y^2)$ is non-increasing for $\alpha \in (-1, T-1]$. Since $f(x) = g(\|x\|_2^2)$ depends only on $\|x\|_2$ and is non-increasing in the radius, each superlevel set $A_t$ is a (possibly empty) Euclidean ball centered at the origin: there exists a radius $\rho(t) \in [0,\infty]$ such that $A_t = B(0, \rho(t))$. Likewise, $A_{t/c} = B(0, \rho(t/c))$. Therefore, for each $t > 0$,

$$\lambda\big(A_t \cap (\mu + A_{t/c})\big) = \lambda\big(B(0, \rho(t)) \cap B(\mu, \rho(t/c))\big).$$

A standard geometric fact is that, overlap of two balls decreases as their center separate. Formally, for fixed radii $r_1, r_2 \geq 0$, the function

$$\lambda\big(B(0, r_1) \cap B(se_1, r_2)\big)$$

is non-increasing in $s \geq 0$, where $e_1 = \frac{\mu}{\|\mu\|_2}$ is a fixed reference direction. Since $\lambda(B(0,r_1) \cap B(\mu, r_2))$ depends on $\mu$ only through $s = \|\mu\|_2$, it follows that $\lambda\big(A_t \cap (\mu + A_{t/c})\big)$ is non-increasing in $s = \|\mu\|_2$ for every $t$. Plugging this shows that the overlap integral $\int \min\{f(x), cf(x-\mu)\}\,\mathrm{d}x$ is non-increasing in $s$.

Finally, $\delta_\mu^*(\varepsilon)$ is therefore non-decreasing in $s = \|\mu\|_2$. Hence, if $\|\mu_1\|_2 \leq \|\mu_2\|_2$, then $\delta_{\mu_1}^*(\varepsilon) \leq \delta_{\mu_2}^*(\varepsilon)$, as claimed. $\qquad\square$

---

**Algorithm 1** $\eta$-tight upper-bound oracle $\mathsf{Oracle}_{\mathsf{SGG}}$

---

1: **Input:** Parameters $(T, \alpha, p, \beta, \varepsilon)$ with $\alpha \in (-1, T-1]$, $p > 0$, $\beta > 0$, sensitivity bound $s \geq 0$, target slack $\eta \geq 0$.
2: **Output:** $\widehat{\delta} = \mathsf{Oracle}_{\mathsf{SGG}}(\alpha, p, \beta, \varepsilon, s)$.
3: $k \leftarrow (\alpha + 1)/p$, $\quad \eta_{\mathsf{tail}} \leftarrow \eta/3$, $\quad \eta_{\mathsf{int}} \leftarrow \eta - \eta_{\mathsf{tail}}$
4: Compute (by bisection) any $z_{\max} > 0$ such that $(1 + e^\varepsilon) \, Q(k, z_{\max}) \leq \eta_{\mathsf{tail}}$.

   *Adaptive probability binning on $[0, z_{\max}]$ to bracket the truncated integrals.*
5: $\mathcal{P} \leftarrow \{[0, z_{\max}]\}$                                                        *// current partition into bins*
6: **while true do**
7:      *// For each bin $I_i = [z_i, z_{i+1}]$, compute its Gamma mass and a binwise bracket on $\widetilde{g}_\pm$.*
8:      Initialize accumulators: $\underline{A} \leftarrow 0$, $\overline{A} \leftarrow 0$, $\underline{B} \leftarrow 0$, $\overline{B} \leftarrow 0$.
9:      Initialize bin-score list (for refinement): $\mathrm{score}(I) \leftarrow 0$ for all $I \in \mathcal{P}$.
10:      **for each** $I_i = [z_i, z_{i+1}] \in \mathcal{P}$ **do**
11:          $\pi_i \leftarrow P(k, z_{i+1}) - P(k, z_i)$                          *//P denotes regularized lower incomplete gamma function*
12:          $[\underline{g}_{-,i}, \overline{g}_{-,i}] \leftarrow \mathsf{EncloseG}(I_i, -\varepsilon; T, \alpha, p, \beta, \varepsilon, s)$
13:          $[\underline{g}_{+,i}, \overline{g}_{+,i}] \leftarrow \mathsf{EncloseG}(I_i, \varepsilon; T, \alpha, p, \beta, \varepsilon, s)$
14:          $\underline{A} \leftarrow \underline{A} + \pi_i \, \underline{g}_{-,i}$; $\quad \overline{A} \leftarrow \overline{A} + \pi_i \, \overline{g}_{-,i}$
15:          $\underline{B} \leftarrow \underline{B} + \pi_i \, \underline{g}_{+,i}$; $\quad \overline{B} \leftarrow \overline{B} + \pi_i \, \overline{g}_{+,i}$
16:          $\mathrm{score}(I_i) \leftarrow \pi_i(\overline{g}_{-,i} - \underline{g}_{-,i}) + e^\varepsilon \pi_i(\overline{g}_{+,i} - \underline{g}_{+,i})$
17:      **end for**
18:      **if** $(\overline{A} - \underline{A}) + e^\varepsilon(\overline{B} - \underline{B}) \leq \eta_{\mathsf{int}}$ **then break**
19:      *// Refine the partition by splitting the bin with largest contribution to the bracket width.*
20:      Let $I^\star = [z_L, z_U] \in \mathcal{P}$ be a bin maximizing $\mathrm{score}(I)$.
21:      $z_M \leftarrow (z_L + z_U)/2$.
22:      $\mathcal{P} \leftarrow (\mathcal{P} \setminus \{I^\star\}) \cup \{[z_L, z_M], [z_M, z_U]\}$.
23: **end while**

   *// Add the tail bound and output an upper bound on $\delta_\mu^*(\varepsilon)$.*
24: $\tau \leftarrow Q(k, z_{\max})$                                         *// Q denotes regularized upper incomplete gamma function*
25: $\widehat{\delta} \leftarrow \max\{0, \, (\overline{A} + \tau) - e^\varepsilon \underline{B}\}$
26: **Output:** $\widehat{\delta}$.

---

## C.2. Provable DP Guarantee and Mechanism

We first provide intuition for the proof of Lemma 4.3 (existence of an $\eta$-tight upper-bound oracle $\mathsf{Oracle}_{\mathsf{SGG}}$). It follows by an explicit construction. At a high level, the oracle reduces the one-dimensional integrals defining $\delta_\mu^*(\varepsilon)$ to expectations under a standard $\Gamma(k, 1)$ measure via the change of variables $Z = \beta R^p$ (with $k = (\alpha + 1)/p$), so that Gamma tail probabilities can be computed in closed form. It then (i) truncates the integral at a data-independent threshold $z_{\max}$ chosen to make the discarded tail contribute at most $\eta_{\mathsf{tail}}$, and (ii) brackets the remaining truncated expectations by an adaptive probability-binning scheme: on each bin, the integrand is provably upper and lower bounded by an an interval $[\underline{g}, \overline{g}]$, and these bounds are aggregated with the bin masses $\pi_i$ to obtain global lower/upper bounds whose total width is provably controlled. We now give the formal proof.

*Proof of Lemma 4.3.* We prove the lemma constructively by showing that Algorithm 1 implements an $\eta$-tight upper-bound oracle defined in Definition 4.3.

Fix any dimension $T \geq 2$, parameters $\alpha \in (-1, T-1]$, $p > 0$, $\beta > 0$, privacy level $\varepsilon \geq 0$, and any shift vector $\mu \in \mathbb{R}^T$ with $\|\mu\|_2 = s$. Let $X \sim \mathsf{SGG}(\alpha, \beta, p)$. By Lemma 4.1, the optimal $\delta$ for the neighboring pair $(X, X + \mu)$ can be written as

$$\delta_\mu^*(\varepsilon) = \max\{0, A - e^\varepsilon B\}, \tag{15}$$

where $A = \mathbf{Ex}\left[g_-(R)\right]$ and $B = \mathbf{Ex}\left[g_+(R)\right]$ are expectations under $R \sim \mathsf{GGamma}(\alpha, \beta, p)$, and $g_\pm(\cdot) \in [0, 1]$ are the

---

**Algorithm 2** ENCLOSEG: Binwise bounds on $\widetilde{g}_\pm(z)$ over $I = [z_L, z_U]$

---

1: **Input:** Bin $I = [z_L, z_U]$, target $y \in \{\varepsilon, -\varepsilon\}$, parameters $(T, \alpha, p, \beta, \varepsilon)$, sensitivity bound $s$
2: **Parameter:** bisection tolerance $\tau_w > 0$.
3: **Output:** Bounds $[\underline{g}, \overline{g}]$ such that $\widetilde{g}(z) \in [\underline{g}, \overline{g}]$ for all $z \in I$.
4: $r_L \leftarrow (z_L/\beta)^{1/p}, \quad r_U \leftarrow (z_U/\beta)^{1/p}, \quad w^{\mathsf{lo}} \leftarrow -1, \quad w^{\mathsf{hi}} \leftarrow 1, \quad \mathsf{ok} \leftarrow \mathsf{true}$
     // *Interval-bisection in $w$ for* $\Phi(r, w) = \frac{\alpha+1-T}{2} \ln\left(1 + \frac{2sw}{r} + \frac{s^2}{r^2}\right) + \beta\left[r^p - (r^2 + 2swr + s^2)^{p/2}\right]$ *on* $r \in [r_L, r_U]$.
5: **while** $w^{\mathsf{hi}} - w^{\mathsf{lo}} > \tau_w$ **do**
6:      $w^{\mathsf{mid}} \leftarrow (w^{\mathsf{lo}} + w^{\mathsf{hi}})/2$.
7:      $[\underline{\Phi}, \overline{\Phi}] \leftarrow \mathsf{EnclosePhi}\left([r_L, r_U], w^{\mathsf{mid}}; T, \alpha, p, \beta, s\right)$.
8:      **if** $\underline{\Phi} \leq y \leq \overline{\Phi}$ **then** $\mathsf{ok} \leftarrow \mathsf{false}$; **break** // *Cannot decide the sign of $\Phi(r, w^{\mathsf{mid}}) - y$ uniformly on this bin within $\tau_w$.*
9:      **if** $\underline{\Phi} > y$ **then** $w^{\mathsf{lo}} \leftarrow w^{\mathsf{mid}}$ **else** $w^{\mathsf{hi}} \leftarrow w^{\mathsf{mid}}$
10: **end while**
11: **if** $\mathsf{ok} = \mathsf{false}$ **then Return:** $[0, 1]$
12: **if** $y = \varepsilon$ **then** $\underline{g} \leftarrow F_W(w^{\mathsf{lo}}), \quad \overline{g} \leftarrow F_W(w^{\mathsf{hi}}) \quad$ **else** $\underline{g} \leftarrow 1 - F_W(w^{\mathsf{hi}}), \quad \overline{g} \leftarrow 1 - F_W(w^{\mathsf{lo}})$
13: **Output:** $[\underline{g}, \overline{g}]$.

---

bounded angle-probability functions defined in Lemma 4.1, i.e.

$$g_-(R) = 1 - F_W(w^*(R, -\varepsilon)), \quad g_+(R) = F_W(w^*(R, \varepsilon)).$$

We now show that Algorithm 1 outputs a value $\widehat{\delta}$ satisfying

$$\delta_\mu^*(\varepsilon) \leq \widehat{\delta} \leq \delta_\mu^*(\varepsilon) + \eta. \tag{16}$$

**Rewriting Under a Standard Gamma Measure.** Let $k = (\alpha+1)/p$ and define the change of variables $Z = \beta R^p$. Then $Z \sim \mathrm{Gamma}(k, 1)$ (shape $k$, unit scale), with density

$$f_Z(z) = \frac{1}{\Gamma(k)} z^{k-1} e^{-z}, \qquad z > 0.$$

Define $\widetilde{g}_\pm(z) \in [0, 1]$ by $\widetilde{g}_\pm(z) = g_\pm\left(\left(\frac{z}{\beta}\right)^{1/p}\right)$. A direct substitution yields

$$A = \mathbf{Ex}\left[\widetilde{g}_-(Z)\right], \quad B = \mathbf{Ex}\left[\widetilde{g}_+(Z)\right].$$

**Truncation and a Closed-form Tail Bound.** Let $z_{\max} > 0$ and set $\tau = \Pr[Z > z_{\max}] = Q(k, z_{\max})$, where $Q(\cdot, \cdot)$ is the regularized upper incomplete gamma function. Define the truncated integrals

$$A_{\leq z_{\max}} = \int_0^{z_{\max}} f_Z(z) \widetilde{g}_-(z) \, dz, \quad B_{\leq z_{\max}} = \int_0^{z_{\max}} f_Z(z) \widetilde{g}_+(z) \, dz.$$

Since $0 \leq \widetilde{g}_\pm \leq 1$, the discarded tails satisfy $0 \leq A - A_{\leq z_{\max}} \leq \tau, \quad 0 \leq B - B_{\leq z_{\max}} \leq \tau$. Consequently,

$$\left|(A - e^\varepsilon B) - (A_{\leq z_{\max}} - e^\varepsilon B_{\leq z_{\max}})\right| \leq (1 + e^\varepsilon)\tau.$$

In Algorithm 1, we choose $z_{\max}$ (by bisection on $Q(k, \cdot)$) so that $(1 + e^\varepsilon)\tau \leq \eta_{\mathrm{tail}}$. This guarantees that truncation contributes at most $\eta_{\mathrm{tail}}$ additive error to the linear form $A - e^\varepsilon B$.

**Probability binning and Brackets on the Truncated Integrals.** Fix any partition $0 = z_0 < z_1 < \cdots < z_m = z_{\max}, \quad I_i = [z_i, z_{i+1}]$. Let $\pi_i = \Pr[Z \in I_i] = P(k, z_{i+1}) - P(k, z_i)$, where $P(\cdot, \cdot)$ is the regularized lower incomplete gamma function. Suppose that for each bin $I_i$ we have a provable lower and upper bound

$$\widetilde{g}_\pm(z) \in [\underline{g}_{\pm,i}, \overline{g}_{\pm,i}], \quad \forall z \in I_i.$$

---

**Algorithm 3** ENCLOSEPHI: Interval bound for $\Phi(r, w)$ over $r \in [r_L, r_U]$

---

1: **Input:** interval $[r_L, r_U]$ with $0 < r_L \leq r_U$, scalar $w \in [-1, 1]$, parameters $(T, \alpha, p, \beta, s)$
2: **Output:** $[\underline{\Phi}, \overline{\Phi}]$ s.t. $\Phi(r, w) \in [\underline{\Phi}, \overline{\Phi}]$ for all $r \in [r_L, r_U]$.
3: $c_1 \leftarrow (\alpha + 1 - T)/2$
   // *Bound $q(r) = r^2 + 2swr + s^2$ on $[r_L, r_U]$ (convex quadratic).*
4: $r^\star \leftarrow \min\{r_U, \max\{r_L, -sw\}\}$ {projection of the vertex $-sw$}
5: $q_L \leftarrow r_L^2 + 2swr_L + s^2, \quad q_U \leftarrow r_U^2 + 2swr_U + s^2, \quad q_\star \leftarrow (r^\star)^2 + 2swr^\star + s^2$
6: $\underline{q} \leftarrow \min\{q_L, q_U, q_\star\}, \quad \overline{q} \leftarrow \max\{q_L, q_U\}$
   // *Bound $c_1 \ln(q(r)/r^2)$ using $\underline{q}, \overline{q}$ and $r^2 \in [r_L^2, r_U^2]$.*
7: $\underline{u} \leftarrow \underline{q}/r_U^2, \quad \overline{u} \leftarrow \overline{q}/r_L^2$
8: **if** $\underline{u} \leq 0$ **then** $\underline{\ell} \leftarrow -\infty$ **else** $\underline{\ell} \leftarrow \ln(\underline{u})$
9: $\overline{\ell} \leftarrow \ln(\overline{u})$
10: **if** $c_1 \geq 0$ **then** $\underline{L} \leftarrow c_1\underline{\ell}, \quad \overline{L} \leftarrow c_1\overline{\ell}$ **else** $\underline{L} \leftarrow c_1\overline{\ell}, \quad \overline{L} \leftarrow c_1\underline{\ell}$
   // *Bound $r^p - q(r)^{p/2}$ (monotone for $r \geq 0, q \geq 0$).*
11: $\underline{a} \leftarrow r_L^p, \quad \overline{a} \leftarrow r_U^p$
12: $\underline{b} \leftarrow (\underline{q})^{p/2}, \quad \overline{b} \leftarrow (\overline{q})^{p/2}$
13: $\underline{D} \leftarrow \underline{a} - \overline{b}, \quad \overline{D} \leftarrow \overline{a} - \underline{b}$
   // *Combine: $\Phi = L + \beta \cdot D$ (with $\beta > 0$).*
14: $\underline{\Phi} \leftarrow \underline{L} + \beta \underline{D}$
15: $\overline{\Phi} \leftarrow \overline{L} + \beta \overline{D}$
16: **Output:** $[\underline{\Phi}, \overline{\Phi}]$.

---

Multiplying by the nonnegative density and integrating over $I_i$ yields

$$\underline{A} = \sum_{i=0}^{m-1} \pi_i \, \underline{g}_{-,i} \leq A_{\leq z_{\max}} \leq \sum_{i=0}^{m-1} \pi_i \, \overline{g}_{-,i} = \overline{A},$$

$$\underline{B} = \sum_{i=0}^{m-1} \pi_i \, \underline{g}_{+,i} \leq B_{\leq z_{\max}} \leq \sum_{i=0}^{m-1} \pi_i \, \overline{g}_{+,i} = \overline{B}.$$

Therefore the truncated linear form is bracketed as

$$\underline{A} - e^\varepsilon \overline{B} \leq A_{\leq z_{\max}} - e^\varepsilon B_{\leq z_{\max}} \leq \overline{A} - e^\varepsilon \underline{B},$$

and the bracket width satisfies

$$\left(\overline{A} - \underline{A}\right) + e^\varepsilon \left(\overline{B} - \underline{B}\right) = \sum_{i=0}^{m-1} \pi_i \left[\left(\overline{g}_{-,i} - \underline{g}_{-,i}\right) + e^\varepsilon (\overline{g}_{+,i} - \underline{g}_{+,i})\right],$$

which is exactly the quantity tracked by line 18 in Algorithm 1.

It remains to justify that $\widetilde{g}_\pm(z) \in [\underline{g}_{\pm,i}, \overline{g}_{\pm,i}]$ is true for all $i \in [m], z \in I_i$, and that the adaptive refinement loop terminates once $\left(\overline{A} - \underline{A}\right) + e^\varepsilon \left(\overline{B} - \underline{B}\right)$ falls below $\eta_{\mathrm{int}}$.

**Correctness of ENCLOSEPHI and ENCLOSEG.** We first recall the (binwise) privacy-loss threshold function used in Lemma 4.1. For $r > 0$ and $w \in [-1, 1]$, define

$$\Phi(r, w) = \frac{\alpha + 1 - T}{2} \ln\left(1 + \frac{2sw}{r} + \frac{s^2}{r^2}\right) + \beta \left[r^p - (r^2 + 2swr + s^2)^{p/2}\right].$$

Because $\alpha \leq T - 1$, the coefficient $(\alpha + 1 - T)/2 \leq 0$, and since $q(r, w) = r^2 + 2swr + s^2$ increases with $w$, both terms in $\Phi(r, w)$ are non-increasing in $w$. Hence, for each fixed $r > 0$, the map $w \mapsto \Phi(r, w)$ is non-increasing on $[-1, 1]$.

For a target level $y \in \{\varepsilon, -\varepsilon\}$, define the threshold

$$w^*(r, y) = \sup\{w \in [-1, 1] : \Phi(r, w) \geq y\},$$

with the convention that $\sup \varnothing = -1$. By monotonicity in $w$, we have that for every $r > 0$,

$$\Phi(r, w) \geq y \iff w \leq w^*(r, y).$$

Recall that $g_-(r) = 1 - F_W(w^*(r, -\varepsilon))$, $\quad g_+(R) = F_W(w^*(r, \varepsilon))$, and $\widetilde{g}_\pm(z) = g_\pm\left(\left(\frac{z}{\beta}\right)^{1/p}\right)$.

*Correctness of* ENCLOSEPHI. Fix $w \in [-1, 1]$ and an interval $r \in [r_L, r_U]$ with $0 < r_L \leq r_U$. Algorithm 3 computes bounds for $\Phi(r, w)$ by interval arithmetic: (1) it bounds the quadratic $q(r, w) = r^2 + 2swr + s^2$ on $[r_L, r_U]$ using convexity, (2) it bounds $\ln(q(r, w)/r^2)$ using monotonicity of $\ln$ together with $q(r, w) \in [\underline{q}, \overline{q}]$ and $r^2 \in [r_L^2, r_U^2]$, and (3) it bounds $r^p - q(r, w)^{p/2}$ using monotonicity of $x \mapsto x^{p/2}$ on $\mathbb{R}_{\geq 0}$ and $r^p \in [r_L^p, r_U^p]$. Combining these bounds linearly yields an interval $[\underline{\Phi}, \overline{\Phi}]$ satisfying

$$\Phi(r, w) \in [\underline{\Phi}, \overline{\Phi}] \qquad \forall r \in [r_L, r_U].$$

*Correctness of* ENCLOSEG. Fix a $z$-bin $I = [z_L, z_U]$ and let $r \in [r_L, r_U]$ be its corresponding $r$-range with $r = (z/\beta)^{1/p}$. WLOG, assume now $0 < r_L \leq r_U$. Algorithm 2 performs bisection on $w \in [-1, 1]$ to enclose $w^*(r, y)$ for all $r \in [r_L, r_U]$. At each bisection step with midpoint $w^{\mathrm{mid}}$, it calls ENCLOSEPHI to obtain an interval $[\underline{\Phi}, \overline{\Phi}]$ satisfying $\Phi(r, w^{\mathrm{mid}}) \in [\underline{\Phi}, \overline{\Phi}]$. If $\underline{\Phi} > y$, then $\Phi(r, w^{\mathrm{mid}}) > y$ for all $r$ in the bin; since $\Phi(r, \cdot)$ is non-increasing, this implies $w^*(r, y) \geq w^{\mathrm{mid}}$ for all $r$, so updating $w^{\mathrm{lo}} \leftarrow w^{\mathrm{mid}}$ is sound. Similarly, if $\overline{\Phi} < y$. If neither condition holds (i.e., the interval straddles $y$), the routine returns $[0, 1]$, which again is sound.

When the routine terminates without the fallback, it has produced $w^{\mathrm{lo}} \leq w^{\mathrm{hi}}$ such that

$$w^*(r, y) \in [w^{\mathrm{lo}}, w^{\mathrm{hi}}], \quad \forall r \in [r_L, r_U].$$

Applying the monotonicity of the CDF $F_W$ yields

$$F_W(w^*(r, \varepsilon)) \in \big[F_W(w^{\mathrm{lo}}), F_W(w^{\mathrm{hi}})\big], \quad , 1 - F_W(w^*(r, -\varepsilon)) \in \big[1 - F_W(w^{\mathrm{hi}}), 1 - F_W(w^{\mathrm{lo}})\big],$$

which is exactly what Algorithm 2 outputs. Composing with $r = (z/\beta)^{1/p}$, we justify that $\widetilde{g}_\pm(z) \in [\underline{g}_{\pm, i}, \overline{g}_{\pm, i}]$ is true for all $i \in [m], z \in I_i$.

**Termination of the Adaptive Refinement Loop.** We show that the refinement loop in Algorithm 1 terminates, i.e., it eventually constructs a partition such that

$$(\overline{A} - \underline{A}) + e^\varepsilon(\overline{B} - \underline{B}) \leq \eta_{\mathrm{int}}.$$

Over the compact interval $[0, z_{\max}]$, the functions $\widetilde{g}_\pm(z)$ take values in $[0, 1]$ and are Borel-measurable (indeed, they are continuous except possibly at $z = 0$, and bounded everywhere). Hence, for every error bound $\eta > 0$ there exists a partition of $[0, z_{\max}]$ such that the oscillation of $\widetilde{g}_\pm$ on each bin is at most $\eta$, except possibly on bins that touch 0. For bins touching 0, even the trivial enclosure $[0, 1]$ contributes at most their Gamma mass to the error accumulator; since $k > 0$ the Gamma measure of $[0, \zeta]$ tends to 0 as $\zeta \downarrow 0$, so by refining near 0 we can make the total contribution of such bins arbitrarily small. We also note that the accumulated bracket width is a *probability-weighted* sum of binwise oscillations, and hence does *not* scale with the number of bins.

Formally, pick $\eta > 0$ such that $(1 + e^\varepsilon)\eta \leq \eta_{\mathrm{int}}/2$. Choose $\zeta \in (0, z_{\max})$ so that $(1 + e^\varepsilon) \Pr[Z \in [0, \zeta]] \leq \eta_{\mathrm{int}}/2$. On $[\zeta, z_{\max}]$, bounded (and continuous) functions are uniformly continuous, so there exists a finite partition of $[\zeta, z_{\max}]$ in which each bin has oscillation at most $\eta$ for both $\widetilde{g}_-$ and $\widetilde{g}_+$. Using the correctness property of ENCLOSEG on those bins then yields

$$\sum_{I \subseteq [\zeta, z_{\max}]} \pi_I \big[(\overline{g}_{-, I} - \underline{g}_{-, I}) + e^\varepsilon(\overline{g}_{+, I} - \underline{g}_{+, I})\big] \leq (1 + e^\varepsilon)\delta \leq \eta_{\mathrm{int}}/2.$$

For bins contained in $[0, \zeta]$, even the trivial enclosure $[0, 1]$ gives

$$\sum_{I \subseteq [0, \zeta]} \pi_I \big[(\overline{g}_{-, I} - \underline{g}_{-, I}) + e^\varepsilon(\overline{g}_{+, I} - \underline{g}_{+, I})\big] \leq (1 + e^\varepsilon) \Pr[Z \in [0, \zeta]] \leq \eta_{\mathrm{int}}/2.$$

---

**Algorithm 4** Parameterized SGG Mechanism $\mathcal{M}_{\mathsf{SGG}}^{\alpha,p}$

1: **Input:** A vector-valued query $q(G) \in \mathbb{R}^T$ with $\ell_2$ sensitivity $s > 0$, privacy parameters $(\varepsilon, \delta)$, fixed $(\alpha, p)$ with $\alpha \in (-1, T-1]$, $p > 0$, $\eta$-tight upper-bound oracle $\mathsf{Oracle}_{\mathsf{SGG}}$ (Definition 4.3), tolerance $\tau > 0$.
2: $\beta^{\mathsf{lo}} \leftarrow 1$
3: **while** $\mathsf{Oracle}_{\mathsf{SGG}}(\alpha, p, \beta^{\mathsf{lo}}, \varepsilon, s) > \delta$ **do** $\beta^{\mathsf{lo}} \leftarrow \beta^{\mathsf{lo}}/2$
4: $\beta^{\mathsf{hi}} \leftarrow \beta^{\mathsf{lo}}$
5: **while** $\mathsf{Oracle}_{\mathsf{SGG}}(\alpha, p, \beta^{\mathsf{hi}}, \varepsilon, s) \leq \delta$ **do** $\beta^{\mathsf{hi}} \leftarrow 2\beta^{\mathsf{hi}}$
6: **while** $\beta^{\mathsf{hi}}/\beta^{\mathsf{lo}} > 1 + \tau$ **do**
7: $\quad \beta^{\mathsf{mid}} \leftarrow (\beta^{\mathsf{lo}} + \beta^{\mathsf{hi}})/2.$
8: $\quad$ **if** $\mathsf{Oracle}_{\mathsf{SGG}}(\alpha, p, \beta^{\mathsf{mid}}, \varepsilon, s) \leq \delta$ **then** $\beta^{\mathsf{lo}} \leftarrow \beta^{\mathsf{mid}}$ **else** $\beta^{\mathsf{hi}} \leftarrow \beta^{\mathsf{mid}}$
9: **end while**
10: $\beta \leftarrow \beta^{\mathsf{lo}}.$
11: **Output:** $q(G) + X$, where $X \sim \mathsf{SGG}(\alpha, \beta, p)$.

---

Adding the two contributions gives $(\overline{A} - \underline{A}) + e^\varepsilon(\overline{B} - \underline{B}) \leq \eta_{\mathrm{int}}$. Since Algorithm 1 refines by repeatedly splitting the bin with largest contribution to $(\overline{A} - \underline{A}) + e^\varepsilon(\overline{B} - \underline{B})$, its mesh size tends to 0 and it must eventually reach a refinement at least as fine as the existence argument above; therefore it terminates.

**Conclusion.** Let $D = A - e^\varepsilon B$ and let

$$\widehat{D} = (\overline{A} + \tau) - e^\varepsilon \underline{B}, \quad \widehat{\delta} = \max\{0, \widehat{D}\}, \quad \delta_\mu^*(\varepsilon) = \max\{0, D\}.$$

We first show $\widehat{D} \geq D$ (hence $\widehat{\delta} \geq \delta_\mu^*(\varepsilon)$). This is because

$$A = A_{\leq z_{\max}} + (A - A_{\leq z_{\max}}) \leq \overline{A} + \tau, \quad B \geq B_{\leq z_{\max}} \geq \underline{B},$$

and therefore $D = A - e^\varepsilon B \leq (\overline{A} + \tau) - e^\varepsilon \underline{B} = \widehat{D}$.

Next, we upper bound $\widehat{D} - D$. Using the decompositions $A = A_{\leq z_{\max}} + A_{> z_{\max}}$ and $B = B_{\leq z_{\max}} + B_{> z_{\max}}$ with $0 \leq A_{> z_{\max}}, B_{> z_{\max}} \leq \tau$, we have

$$\begin{aligned}
\widehat{D} - D &= (\overline{A} - A_{\leq z_{\max}}) + (\tau - A_{> z_{\max}}) + e^\varepsilon(B_{\leq z_{\max}} - \underline{B}) + e^\varepsilon B_{> z_{\max}} \\
&\leq (\overline{A} - \underline{A}) + e^\varepsilon(\overline{B} - \underline{B}) + (1 + e^\varepsilon)\tau \\
&\leq \eta_{\mathrm{int}} + \eta_{\mathrm{tail}} = \eta.
\end{aligned}$$

Finally, since $x \mapsto \max\{0, x\}$ is 1-Lipschitz, $0 \leq \widehat{\delta} - \delta_\mu^*(\varepsilon) \leq \widehat{D} - D \leq \eta$, which is exactly Equation (16). So we conclude that Algorithm 1 is an $\eta$-tight upper-bound oracle, and such an oracle exists. $\qquad\square$

With Lemma 4.3 in hand, we show some supplementary statements needed for our mechhanism in Algorithm 4.

Lemma C.1 states that, for fixed $(\alpha, p)$, changing the rate parameter $\beta$ simply rescales the SGG noise by a factor $\lambda = \beta^{-1/p}$. Equivalently, $\mathsf{SGG}(\alpha, \beta, p)$ is the distribution of $\lambda X_1$ for $X_1 \sim \mathsf{SGG}(\alpha, 1, p)$.

**Lemma C.1.** *Fix $T \geq 2$, $\alpha > -1$, $p > 0$, and $\beta > 0$, and define $\lambda = \beta^{-1/p}$. Let $R_1 \sim \mathsf{GGamma}(\alpha, 1, p)$ and $U \sim \mathsf{Unif}(\mathcal{S}^{T-1})$ be independent, and let $X_1 \triangleq R_1 U \sim \mathsf{SGG}(\alpha, 1, p)$. Then*

$$R \sim \mathsf{GGamma}(\alpha, \beta, p) \quad \Longleftrightarrow \quad R \overset{d}{=} \lambda R_1,$$

*and consequently,*

$$X \sim \mathsf{SGG}(\alpha, \beta, p) \quad \Longleftrightarrow \quad X \overset{d}{=} \lambda X_1.$$

**Proposition C.1** (Monotonicity in the rate parameter $\beta$). *Fix $T \geq 2$, $\alpha \in (-1, T-1]$, $p > 0$, and $\varepsilon \geq 0$. For $\beta > 0$, let $X_\beta \sim \mathsf{SGG}(\alpha, \beta, p)$ and let $\delta_\mu^*(\varepsilon; \beta)$ denote the optimal delta for the neighboring pair $(X_\beta, X_\beta + \mu)$ at level $\varepsilon$. Then, for any fixed shift $\mu \in \mathbb{R}^T$, the mapping $\beta \mapsto \delta_\mu^*(\varepsilon; \beta)$ is non-decreasing. Equivalently, for $0 < \beta_1 \leq \beta_2$,*

$$\delta_\mu^*(\varepsilon; \beta_1) \leq \delta_\mu^*(\varepsilon; \beta_2).$$

*Proof.* Let $\lambda(\beta) = \beta^{-1/p}$. By Lemma C.1, we may write $X_\beta \overset{d}{=} \lambda(\beta) \, X_1$ where $X_1 \sim \mathsf{SGG}(\alpha, 1, p)$. Consider the invertible map $\varphi_\lambda(x) = x/\lambda$. Since hockey-stick divergence (and hence $\delta_\mu^*(\varepsilon)$) is invariant under bijective transformations, we have

$$
\begin{aligned}
\delta_\mu^*(\varepsilon; \beta) &= \delta^*\big(\varepsilon;\, X_\beta,\, X_\beta + \mu\big) \\
&= \delta^*\big(\varepsilon;\, \varphi_{\lambda(\beta)}(X_\beta),\, \varphi_{\lambda(\beta)}(X_\beta + \mu)\big) \\
&= \delta^*\big(\varepsilon;\, X_1,\, X_1 + \mu/\lambda(\beta)\big) = \delta_{\mu/\lambda(\beta)}^*(\varepsilon; 1).
\end{aligned}
$$

Now fix $0 < \beta_1 \le \beta_2$. Then $\lambda(\beta_1) \ge \lambda(\beta_2)$, hence

$$
\left\| \frac{\mu}{\lambda(\beta_1)} \right\|_2 \le \left\| \frac{\mu}{\lambda(\beta_2)} \right\|_2.
$$

Applying the shift monotonicity result Lemma 4.2 to the base distribution $X_1 \sim \mathsf{SGG}(\alpha, 1, p)$ yields

$$
\delta_{\mu/\lambda(\beta_1)}^*(\varepsilon; 1) \le \delta_{\mu/\lambda(\beta_2)}^*(\varepsilon; 1).
$$

Substituting back proves $\delta_\mu^*(\varepsilon; \beta_1) \le \delta_\mu^*(\varepsilon; \beta_2)$. $\qquad\square$

---

**Algorithm 5** Spherical Generalized Gamma Mechanism

---

1: **Input:**
- $T \in \mathcal{N}_{>0}$, the query dimension, $s > 0$, the query's $\ell_2$ sensitivity
- $\varepsilon > 0, \delta \in [0, 1]$, the desired privacy parameters
- $q(G) \in \mathbb{R}^T$, query result
- $\mathsf{Oracle}_{\mathsf{SGG}}$ which on input $T, \alpha, p, \beta, \varepsilon$ outputs an upper bound for $\delta$.
- $\mathsf{err} : [0, T-1] \times \mathbb{R}_{>0} \times \mathbb{R}_{>0} \mapsto \mathbb{R}_{>0}$, the error function for generalized gamma distribution

2: **Parameters:**
- $\alpha \in [0, T-1], \beta > 0, p > 0$ the parameters of the generalized gamma distribution

3: Set the noise budget upper bound $c^{\mathsf{upp}}$ such that the standard Gaussian mechanism that achieves $(\varepsilon, \delta)$-DP introduces error $c^{\mathsf{upp}}$. Also set $c^{\mathsf{mid}} \leftarrow 1$.

4: $\delta^* \leftarrow \infty$

5: *// Perform a binary search over $c \in [1, c^{\mathsf{upp}}]$ to find the smallest feasible noise level*

6: **while** $\delta^* > \delta \vee (|\delta^* - \delta| > \mathsf{atol} \wedge |c^{\mathsf{low}} - c^{\mathsf{upp}}| > \mathsf{atol})$ **do**

7:     **if** $\delta^* \le \delta$ **then** set $c^{\mathsf{upp}} \leftarrow c^{\mathsf{mid}}$ **else** set $c^{\mathsf{low}} \leftarrow c^{\mathsf{mid}}$

8:     $c^{\mathsf{mid}} \leftarrow (c^{\mathsf{low}} + c^{\mathsf{upp}})/2$

9:     Solve the 2-dimensional optimization problem

$$
\alpha^*, p^* \leftarrow \arg\min_{\alpha, p} \mathsf{Oracle}_{\mathsf{SGG}}(T, \alpha, \beta, p, \varepsilon)
$$

    *// $\beta$ and $\beta^*$ below computed via Algorithm 4 on input $s, \varepsilon, \alpha, p$ and target $\hat{\delta}$, constrained to noise budget $c^{\mathsf{mid}}$*

10:     $\delta^* \leftarrow \mathsf{Oracle}_{\mathsf{SGG}}(T, \alpha^*, \beta^*, p^*, \varepsilon)$

11: **end while**

12: **Output**: $q(G) + RU$, where $R \sim \mathsf{GGamma}\big(\alpha^*, h(\alpha^*, p^*, c^{\mathsf{mid}}), p^*\big)$ and $U \sim \mathsf{Unif}(\mathcal{S}^{T-1})$.

---

**Theorem C.1.** *The mechanism $\mathcal{M}_{\mathsf{SGG}}^{\alpha, p}$ (Algorithm 4) is $(\varepsilon, \delta)$-differentially private.*

*Proof.* Fix neighboring databases $G \sim G'$. Let $\mu = q(G) - q(G') \in \mathbb{R}^T$, and we know $\|\mu\|_2 \le s$ by the assumption on $q(\cdot)$. The mechanism in Algorithm 4 outputs

$$
\mathcal{M}_{\mathsf{SGG}}^{\alpha, p}(G) = q(G) + X_\beta, \qquad \mathcal{M}_{\mathsf{SGG}}^{\alpha, p}(G') = q(G') + X_\beta,
$$

where $X_\beta \sim \mathsf{SGG}(\alpha, \beta, p)$ and $\beta$ is the value returned by the bracketing-and-bisection procedure.

By translation invariance, distinguishing $\mathcal{M}_{\mathsf{SGG}}^{\alpha,p}(G)$ from $\mathcal{M}_{\mathsf{SGG}}^{\alpha,p}(G')$ is equivalent to distinguishing $X_\beta$ from $X_\beta + \mu$. Thus, the optimal $\delta$ at level $\varepsilon$ for the pair of the mechanism outputs equals $\delta_\mu^*(\varepsilon;\beta)$, the optimal $\delta(\varepsilon)$ between $X_\beta$ and $X_\beta + \mu$.

By the shift monotonicity result Lemma 4.2, the map $\mu \mapsto \delta_\mu^*(\varepsilon;\beta)$ is non-decreasing in $\|\mu\|_2$, hence for any $\mu$ with $\|\mu\|_2 \le s$ and any fixed $\beta$,

$$\delta_\mu^*(\varepsilon;\beta) \le \delta_{\mu_s}^*(\varepsilon;\beta) \quad \text{for any } \mu_s \in \mathbb{R}^T \text{ with } \|\mu_s\|_2 = s.$$

Applying the oracle guarantee in Definition 4.3 to $\mu_s$ gives

$$\delta_{\mu_s}^*(\varepsilon;\beta) \le \mathsf{Oracle}_{\mathsf{SGG}}(\alpha,p,\beta,\varepsilon,\|\mu_s\|_2) = \mathsf{Oracle}_{\mathsf{SGG}}(\alpha,p,\beta,\varepsilon,s).$$

**The chosen $\beta$ is feasible for the target $(\varepsilon,\delta)$.** By construction, the algorithm maintains a feasible lower bracket $\beta^{\mathsf{lo}}$ with $\mathsf{Oracle}_{\mathsf{SGG}}(\alpha,p,\beta^{\mathsf{lo}},\varepsilon,s) \le \delta$ and an infeasible upper bracket $\beta^{\mathsf{hi}}$ with $\mathsf{Oracle}_{\mathsf{SGG}}(\alpha,p,\beta^{\mathsf{hi}},\varepsilon,s) > \delta$. The update rule in the bisection step relies on the monotonicity in $\beta$ (Proposition C.1) to preserve feasibility/infeasibility of the brackets. At termination the algorithm sets $\beta \leftarrow \beta^{\mathsf{lo}}$, and therefore

$$\mathsf{Oracle}_{\mathsf{SGG}}(\alpha,p,\beta,\varepsilon,s) \le \delta.$$

Combining the last two inequalities, we conclude that for every neighboring $G \sim G'$, the output distributions satisfy the $(\varepsilon,\delta)$-DP inequality. Formally,

$$\delta_\mu^*(\varepsilon;\beta) \le \mathsf{Oracle}_{\mathsf{SGG}}(\alpha,p,\beta,\varepsilon,s) \le \delta,$$

i.e., $\mathcal{M}_{\mathsf{SGG}}^{\alpha,p}$ is $(\varepsilon,\delta)$-differentially private. $\qquad\square$

### C.3. Finding Optimal Parameters of the SGG Mechanism

Finally, based on Algorithm 4 that finds best $\beta$ for fixed $(\varepsilon,\delta,\alpha,p)$, we provide Algorithm 5, which finds the parameters $(\alpha^*,\beta^*,p^*)$ with minimal MSE given target privacy $(\varepsilon,\delta)$.

### C.4. Finding Settings where SGG Dominates

Here we present Algorithm 6, which for a given $T,s,\varepsilon$, finds the $\delta$ in which the SGG has the biggest advantage over both the Gaussian and $\ell_2$ mechanisms.

---

**Algorithm 6** Finding Optimal Advantage

---

1: **Input:** $T \in \mathcal{N}_{>0}$, the query dimension, $s > 0$, the query's $\ell_2$ sensitivity, and $\varepsilon > 0$, the desired privacy parameter.
2: Set $\delta^{\mathsf{low}} \leftarrow 0, \delta^{\mathsf{upp}} \leftarrow 0.1$.
3: // *Perform a binary search over $\delta \in [\delta^{\mathsf{low}}, \delta^{\mathsf{upp}}]$ to find the largest advantage*
4: **while** $\delta^{\mathsf{upp}} - \delta^{\mathsf{low}} > \mathrm{atol}$ **do**
5: $\quad \delta^{\mathsf{mid}} \leftarrow (\delta^{\mathsf{low}} + \delta^{\mathsf{upp}})/2$
6: $\quad$ Use Algorithm 5 to find $(\alpha^*,\beta^*,p^*)$ achieving minimal MSE $c^*$ of the SGG for $T$-dimensional query with sensitivity $s$ and privacy parameters $(\varepsilon,\delta)$.
7: $\quad$ Compute MSE $c_{\ell_2}$ and $c_G$ of $\ell_2$ and Gaussian mechanisms, respectively for the above setting.
8: $\quad$ **if** $(c_G - c^*)/c_G < (c_{\ell_2} - c^*)/c_{\ell_2}$ **then** set $\delta^{\mathsf{upp}} \leftarrow \delta^{\mathsf{mid}}$ **else** set $\delta^{\mathsf{low}} \leftarrow \delta^{\mathsf{mid}}$
9: **end while**
10: **Output:** $(\alpha^*,\beta^*,p^*),c^*$.

---

## D. Privacy Accounting of SGG Mechanisms

In this section, we study tight composed privacy for SGG mechanisms under multiple invocations.

Our approach follows the FFT-based accountant developed by (Gopi et al., 2024). Their work casts privacy accounting in terms of the *privacy-loss random variable* (PRV). The central benefit is that privacy losses add: the PRV of a $k$-fold

---

**Algorithm 7** Privacy Accounting for $\mathcal{M}_{\mathsf{SGG}}^{\alpha,p}$ under $k$ invocations

---

1: **Input:** Dimension $T$, sensitivity $s$, SGG parameters $(\alpha, \beta, p)$, composition count $k$, target $\varepsilon$
2: **Discretization parameters:** truncation $L > 0$, step size $h > 0$, odd grid size $M = 1 + \lceil 2L/h \rceil$
3: **Numerical parameters:** radial quadrature size $K$, angular grid size $N_w$, arithmetic precision prec
   // Discretize the single-step PRV
4: Define grid centers $z_i \leftarrow -L + ih$ for $i = 0, 1, \ldots, M-1$ and bin edges $b_i \leftarrow z_i - \frac{h}{2}$.
5: For each edge $b_i$, approximate $\Pr[Z \leq b_i]$ using a $K$-node Gauss–Laguerre quadrature (cf. Equation (17)).
6: Set the pmf $\pi_i \leftarrow \Pr[Z \leq b_{i+1}] - \Pr[Z \leq b_i]$
   // Compose by FFT convolution
7: Compute the $k$-fold convolution $\pi^{(*k)}$ using FFTs, cropping back to the support $[-L, L]$ after each multiplication (per (Gopi et al., 2024))
   // Output the composed privacy guarantee
8: Compute $\delta_k(\varepsilon) \leftarrow \sum_{i=0}^{M-1} \pi_i^{(*k)} \cdot \left(1 - e^{\varepsilon - z_i}\right)_+$
9: Return $(\varepsilon, \delta_k(\varepsilon))$

---

composition is the sum of $k$ single-step PRV. This reduces tight $(\varepsilon, \delta)$ accounting to (i) characterizing the distribution of the single-step PRV and (ii) computing the distribution of its $k$-fold sum via repeated convolution. These two observations allows computing tight composed privacy efficiently by using FFTs after discretizations of single-step PRV. We follow (Gopi et al., 2024) along its theoretical analysis, and tailor the missing SGG-specific component: an efficient discretization of the single-step PRV using the one-dimensional integral reduction from Section 4.1.

To discretize the single-step PRV $Z$ for an SGG mechanism, we use the standard approach of placing a grid on the privacy-loss axis and computing bin masses from CDF differences at grid edges. Thus, efficient discretization amounts to evaluating the PRV CDF $\Pr[Z \leq z]$ at many grid points. Lemma D.1 makes this feasible: it expresses $\Pr[Z \leq z]$ as a one-dimensional raidal integral via the same technique used in Section 4.1.

**Lemma D.1.** *(Proof is below) Adopt the setup and notation of Lemma 4.1. Let $X \sim \mathsf{SGG}(\alpha, \beta, p)$ and define the PRV for the pair $(X, X + \mu)$ by*

$$Z \overset{\text{def}}{=} \log \frac{f_X(X)}{f_X(X + \mu)}.$$

*Then, for every $z \in \mathbb{R}$,*

$$\Pr[Z \leq z] = \int_0^\infty f_R(r) F_W\big(w^\star(r, -z)\big) \mathrm{d}r,$$

*where $f_R$ and $F_W$ are as in Lemma 4.1, and $w^\star(r, \cdot)$ denotes the corresponding threshold map.*

The next step is to increase the efficiency of evaluations of $\Pr[Z \leq z]$ over the discretization grid. The efficiency tricks exploit the independence between random variable $R$ and the monotonicity of $w \mapsto \ell(r, w)$. First, we approximate the one-dimensional radial ($r$) integral by a $K$-node Gauss–Laguerre quadrature after the change of variables $t = \beta r^p$, leading fixed radii $\{r_k\}_{k=1}^K$ and weights $\{\omega_k\}_{k=1}^K$ such that

$$\Pr[Z \leq z] \approx \sum_{k=1}^K \omega_k F_W\big(w^*(r_k, -z)\big). \tag{17}$$

Second, for each quadrature node $r_k$ we precompute a dense lookup table of the strictly decreasing map $w \mapsto \ell(r_k, w)$ on a grid over $w \in [-1, 1]$; thereafter, each threshold $w^*(r_k, -z)$ is computed as a table lookup followed by linear interpolation. Third, we evaluate $F_W(\cdot)$ in vectorized form and complete each CDF query with a single dot product against the cached weights. With these caches, computing $\Pr[Z \leq z]$ costs essentially $O(K)$ floating-point operations per $z$. We also (re)use parallel workers to amortize overhead in practice.

Algorithm 7 describes the full SGG-tailored accounting algorithm. We also note that as the $\ell_2$ mechanism is a special case of SGG (e.g., $(\alpha, p) = (T - 1, 1)$ with an appropriate scale), this yields a practical tight accountant for its composed approximate-DP guarantees as well; more broadly, the same pipeline applies to any $(\alpha, \beta, p)$ and to mixed compositions by convolving the corresponding discretized PRVs.

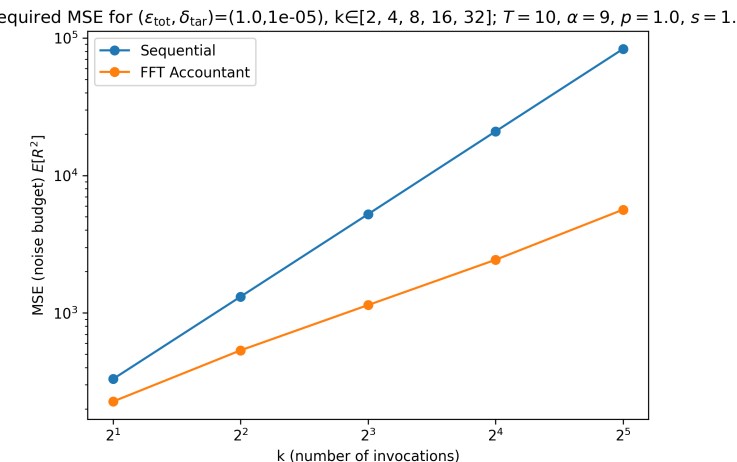

*Figure 3.* MSE Required for privacy target $(\varepsilon_{\text{tot}}, \delta_{\text{tar}}) = (1, 10^{-5})$ under $k$ invocations: sequential composition versus Algorithm 7.

**Experimental Evaluation.**   Figure 3 compares the noise level (measured by the mean-squared error (MSE) of the SGG mechanisms) required to meet a fixed composed privacy target under two accounting methods: (i) the baseline *sequential composition* bound and (ii) our SGG-tailored *FFT accountant* in Algorithm 7. Concretely, we fix a total privacy target $(\varepsilon_{\text{tot}}, \delta_{\text{tar}}) = (1, 10^{-5})$ and consider $k \in \{2, 4, 8, 16, 32\}$ invocations of the same SGG mechanism $\mathcal{M}_{\text{SGG}}^{\alpha,p}$ in dimension $T = 10$ (with fixed shape parameter $\alpha = 9$ and $p = 1$). For each $k$ and for each accountant, we compute the *minimal* per-invocation MSE such that the $k$-fold composition satisfies

$$\delta_k(\varepsilon_{\text{tot}}) \leq \delta_{\text{tar}}.$$

For the sequential baseline, we allocate a per-step budget $(\varepsilon_{\text{tot}}/k, \delta_{\text{tar}}/k)$ and solve for the smallest noise meeting this single-step guarantee (cf. Algorithm 4); for the FFT accountant, we instead solve for the smallest noise such that the computed composed optimal delta from Algorithm 7 meeting $\delta_{\text{tar}}$ at the privacy level $\varepsilon_{\text{tot}}$.

Figure 3 shows a clear and growing gap as $k$ increases between the the baseline accountant and our SGG-tailored FFT accountant. It illustrates the practical impact of tight accounting

*Proof of Lemma D.1.* Write $X = RU$ with $R \sim \mathsf{GGamma}(\alpha, \beta, p)$ independent of $U \sim \mathsf{Unif}(\mathcal{S}^{T-1})$, and let

$$W = \frac{\langle \mu, U \rangle}{\|\mu\|_2} \in [-1, 1].$$

By the spherical form of the SGG density (see the derivation in the proof of Lemma 4.1), the log-density ratio $(-Z)$

$$L^{\text{neg}}(X) = \log \frac{f_X(X + \mu)}{f_X(X)}$$

can be written as a deterministic function of $(R, W)$:

$$L^{\text{neg}}(X) = \ell(R, W),$$

where for each $r > 0$ the map $w \mapsto \ell(r, w)$ is strictly decreasing on $[-1, 1]$ (again, proved in Appendix C.1). Hence, for every $y \in \mathbb{R}$ and $r > 0$, there is a unique threshold $w^*(r, y) \in [-1, 1]$ such that $\ell(r, w^*(r, y)) = y$, and

$$\{\ell(r, W) \geq y\} \iff \{W \leq w^*(r, y)\}.$$

Now note that the PRV in this lemma is the negative of $L^{\text{neg}}$ ($Z = -L^{\text{neg}}(X) = -\ell(R, W)$). Therefore, conditioning on $R = r$ and using the monotonicity property,

$$\Pr[Z \leq z \mid R = r] = \Pr[\ell(r, W) \geq -z] = \Pr[W \leq w^*(r, -z)] = F_W(w^*(r, -z)).$$

| $\varepsilon$ | $\delta_{\text{claimed}}$ | $\psi$ | $\sigma_\star$ | $\delta_{\text{true}}(\varepsilon)$ |
|---|---|---|---|---|
| 0.1 | $10^{-5}$ | 0.798721 | 25.040031 | 0.813284 |
| 1.0 | $10^{-5}$ | 0.798721 | 2.504003 | 0.983594 |
| 2.0 | $10^{-5}$ | 0.798721 | 1.252002 | 0.995020 |
| 4.0 | $10^{-5}$ | 0.798721 | 0.626001 | 0.998804 |
| 8.0 | $10^{-5}$ | 0.798721 | 0.313000 | 0.999755 |

*Table 3.* Testing (Ji & Li, 2024, Theorem 5) calibration for R1SMG at $T = 128$ and $s = 1$ with $\delta_{\text{claimed}} = 10^{-5}$. We compute $\psi$ and the prescribed $\sigma_\star$ from the theorem, then evaluate the resulting privacy $\delta_{\text{true}}(\varepsilon)$ via Lemma 4.1. The computed $\delta_{\text{true}}(\varepsilon)$ is vastly larger than $10^{-5}$, contradicting the claimed guarantee.

Finally, since $R$ is independent of $W$, integrating over $R$ yields

$$\Pr\left[Z \le z\right] = \int_0^\infty f_R(r) \Pr\left[Z \le z \mid R = r\right] \mathrm{d}r = \int_0^\infty f_R(r) F_W\left(w^*(r, -z)\right) \mathrm{d}r,$$

which is exactly the claim. $\qquad\qquad\qquad\qquad\qquad\qquad\qquad\qquad\qquad\qquad\qquad\qquad\qquad\qquad\qquad\qquad\square$

## E. Error in Proof of (Ji & Li, 2024, Theorem 5)

Here we identify an error in the proof of Theorem 5 in (Ji & Li, 2024). The proof attemtps to analyze the Privacy Loss Random Variable (PRV) of their (spherical) R1SMG mechanism. One important fact about the PRV is that it is defined over the randomness of *only* the random variable $\mathcal{M}(q(G))$, and not the randomness of the random variable $\mathcal{M}(q(G'))$ (where $G'$ is an allowed neighbor of $G$). Indeed, consider random variable $Y \stackrel{\text{def}}{=} \mathcal{M}(q(G))$; the PRV is the function

$$L(Y) \stackrel{\text{def}}{=} \ln\left(\frac{f_{\mathcal{M}(q(G))}(Y)}{f_{\mathcal{M}(q(G'))}(Y)}\right),$$

where $f_{\mathcal{M}(q(G))}$ is the density of $\mathcal{M}(q(G))$ and likewise for $f_{\mathcal{M}(q(G'))}$. However, the proof of (Ji & Li, 2024) defines the PRV $L$ as a function of some unspecified vector $s$ and analyzes the PRV over the randomness of both $\mathcal{M}(q(G))$ *and* $\mathcal{M}(q(G'))$. Indeed, they bound the distribution of the angle between the uniformly random directions of the noises used in $\mathcal{M}(q(G))$ and $\mathcal{M}(q(G'))$, and conclude that this angle must be close to $\pi/2$. However, the more appropriate distribution to analyze would be the angle between the noise used in $\mathcal{M}(q(G))$ and the difference vector $\mu \stackrel{\text{def}}{=} q(G) - q(G')$. By similar analysis, it is actually *this* angle that is close to $\pi/2$. This is much different than the angle between the directions in which the noises in $\mathcal{M}(q(G))$ and $\mathcal{M}(q(G'))$ must point to hit $Y$, which the authors seem to attempt to study when considering the ratios of the densities $f_{\mathcal{M}(q(G))}$ and $f_{\mathcal{M}(q(G'))}$ on $Y$; this angle actually can be far from $\pi/2$ since the direction of the noise from $q(G')$ to hit $Y$ will be quite different than the direction of the difference vector $\mu$.

### E.1. Numerical contradiction of the claimed R1SMG calibration

According to (Ji & Li, 2024, Theorem 5), the R1SMG mechanism is $(\varepsilon, \delta)$-DP if its corresponding rank-1 noise parameter $\sigma_\star$ satisfies

$$\psi(T, \delta) = \left(\frac{\delta_{\text{claimed}} \, \Gamma\left(\frac{T-1}{2}\right)}{\sqrt{\pi} \, \Gamma\left(\frac{T}{2}\right)}\right)^{\frac{2}{T-2}},$$

$$\sigma_\star(\varepsilon) \ge \frac{2s^2}{\varepsilon \, \psi(T, \delta_{\text{claimed}})}.$$

We test this claim in dimension $T = 128$ with sensitivity $s = 1$ at target $\delta = 10^{-5}$, for $\varepsilon \in \{0.1, 1, 2, 4, 8\}$. For each $\varepsilon$, we instantiate the R1SMG mechanism using the *minimal* value $\sigma_\star(\varepsilon) = \frac{2s^2}{\varepsilon \, \psi(T, \delta)}$ defined above, and then compute the instantiated mechanism's *actual* optimal delta $\delta_{\text{true}}(\varepsilon)$ via Lemma 4.1. This is because the R1SMG noise is a member of the SGG noise family, so we can use Lemma 4.1 to analyze its privacy. Concretely, the R1SMG noise is rank-1 Gaussian along a random direction; equivalently, it is spherical with half-normal radius $R = \sqrt{\sigma_\star}|Z|$ for $Z \sim \mathcal{N}(0, 1)$. In SGG's

parameterization, this corresponds to an SGG mechanism with parameters

$$\alpha = 0, \qquad p = 2, \qquad \beta = \frac{1}{2\sigma_\star}.$$

Table 3 show a dramatic violation: across $\varepsilon \in \{0.1, 1, 2, 4, 8\}$, the computed $\delta_{\text{true}}(\varepsilon)$ ranges from $0.813$ to $0.9998$, vastly larger than the claimed $\delta = 10^{-5}$.

