# OpenReview forum: "Asymptotic Optimality of the High-Dimensional Gaussian Mechanism and Improved Low-Dimensional Mechanisms for Differential Privacy"
_ICML.cc/2026/Conference — ICML 2026 spotlight_

### Official Review · Reviewer_DC1x · 2026-03-12

**Soundness:** 3
**Presentation:** 3
**Significance:** 3
**Originality:** 3
**Overall Recommendation:** 4
**Confidence:** 3

**Summary:**

The paper presents two theoretical contributions:
1. The first contribution is a new theoretical characterization of the optimality of the Gaussian mechanism
2. The second is the formalization of the Spherical Generalized Gamma (SGG) noise family and its use as an additive DP, providing a theoretical characterization of its benefits over classical constructions (such as Gaussian and Laplace mechanisms).

I find the theoretical contributions in the paper interesting and new, and believe that it presents an important contribution that advances the theory of differential privacy. I have a few questions regarding the presentation of the results and some optimality claims, but overall, I believe that this is a good paper. Thus, I am accepting it.

**Compliance With Llm Reviewing Policy:**

Affirmed.

**Final Justification:**

I thank the authors for the rebuttal response. I feel comfortable with my overall positive assessment, and I maintain my score.

**Key Questions For Authors:**

How would using SGG improve downstream utility measures in settings that currently use Gaussian/$\ell_2$ mechanisms? I believe it would be useful to provide some explanation and theory on the significance of the difference between these two, as it might help users decide when it might be able to improve its algorithms by using SGG.

**Limitations:**

I believe that the paper lacks some experiments showing the significance of the developed theory on some classical tasks in private learning.

**Strengths And Weaknesses:**

The paper provides theoretical contributions regarding the optimality of the Gaussian mechanism in high-dimensional settings and further provides a new class of distributions that might be useful in low-dimensional settings. The results are presented in a clear and comprehensive way, which allows the reader to follow the technical claims. Additionally, I believe that the contributions present a non-trivial extension of the currently existing theory.

I believe that the paper will benefit from more discussion and experiments regarding the significance of the results in downstream applications, showing the benefit of the newly developed theory on tasks in which performance can be measured in a quantitative way. Additionally, given the claims of the usefulness of the SGG in low-dimensional settings, I believe that the paper will benefit from some experiments demonstrating how the theoretical claims improve the performance in settings that currently use the Gaussian mechanism.

---

> ### Author Rebuttal · Authors · 2026-03-31
>
> We thank the reviewer for the thoughtful feedback. We are encouraged that the review generally views the paper as technically sound and recognizes the interest of its theoretical contributions.
>
> ## Downstream utility implications of SGG
> We thank the reviewer for this thoughtful question. We view the low-dimensional SGG contribution primarily as an existence result: in certain concrete low-dimensional regimes, there are better mechanisms than Gaussian and $\ell_2$ mechanisms. We will revise the paper to make this intention clearer, and to be explicit about what the current SGG results do and do not show in practice.
>
> In applications where the additive-noise mechanism itself is the private algorithm, such as low-dimensional vector mean estimation, or vector releases under compositions like the tree mechanism (https://eprint.iacr.org/2010/076.pdf), we expect the gains shown in Fig. 2 to translate into downstream utility improvements in the stated parameter regimes.
>
> By contrast, for private learning settings, such as DP-SGD, additional work is needed to understand whether such low-dimensional gains lead to end-to-end improvements. We believe this stronger claim is not true for high-dimensional DP-SGD, especially since the paper’s asymptotic result supports Gaussian optimality in the high-dimensional regime. We will make this limitation explicit in the revision.

---

> > ### Author Rebuttal · Reviewer_DC1x · 2026-04-03
> >
> > I thank the authors for their answer, which clarified the questions I had. I encourage the authors to include this additional intuition and examples in the final version of the paper.
> >
> > I decide to maintain my overall positive score.

---

> > > ### Author Response · Authors · 2026-04-04
> > >
> > > Thank you for the follow-up and for noting that our response clarified your questions. We appreciate your suggestion and will incorporate the additional intuition and examples into the final version. If you feel that these clarifications strengthen the paper, we would be grateful if you would consider increasing the score.

---

### Official Review · Reviewer_g1PK · 2026-03-12

**Soundness:** 3
**Presentation:** 4
**Significance:** 3
**Originality:** 4
**Overall Recommendation:** 5
**Confidence:** 4

**Summary:**

This paper investigates additive noise mechanisms for real-valued vector queries under the $(\varepsilon,\delta)$ -DP framework. At its core, it addresses two natural and significant questions: First, whether the Gaussian mechanism truly achieves optimality in high dimensions; second, whether a novel mechanism exists that outperforms both the Gaussian mechanism and the recently proposed $\ell_2$ mechanism in low dimensions. The paper's starting point is clear: while the Gaussian mechanism is the most widely used in practice, its theoretical justification for being the default choice has been incomplete, and existing results suggest it may not always be optimal in low dimensions. The first major contribution proves that, under a fixed MSE budget, the Gaussian mechanism is asymptotically optimal over all additive noise mechanisms within a sufficiently strong privacy bound as the dimension $T\to\infty$. The authors' technical approach involves first reducing the problem to a spherically symmetric noise distribution, then transforming the privacy comparison into a one-dimensional analysis of radial energy. This ultimately leads to the conclusion that the Gaussian mechanism cannot be further improved by other additive noise mechanisms in high dimensions. This result is particularly significant for high-dimensional DP learning scenarios, where parameter dimensions in deep model training are often extremely large. The second major contribution is the introduction of a more general family of mechanisms called Spherical Generalized Gamma (SGG). This family encompasses both Gaussian and $\ell_2$ mechanisms. The authors further reduce its privacy analysis to a one-dimensional integral and provide numerical oracle and parameter optimization methods with strict error bounds. Within this framework, the paper identifies mechanisms outperforming both Gaussian and $\ell_2$ mechanisms in several low-dimensional settings, achieving up to 15% improvement in reported MSE. In other words, the paper not only demonstrates that “Gaussian is correct in high dimensions” but also shows that “Gaussian is not necessarily optimal in low dimensions,” providing a systematic alternative design space. Furthermore, the paper fills the gap in tight composition/accounting for the SGG mechanism family. This is not merely an added value but a crucial step determining whether such new mechanisms can truly integrate into existing DP workflows.

**Compliance With Llm Reviewing Policy:**

Affirmed.

**Key Questions For Authors:**

1. On the finite-dimensional relevance of the high-dimensional asymptotic result:
The paper’s main theoretical guarantee shows that, in the strong-privacy regime $\delta \le \delta^\star$, the Gaussian mechanism is asymptotically optimal among all additive-noise mechanisms at fixed MSE as $T \to \infty$. The paper also provides numerical lower bounds on $\delta^\star$, which help show that the regime is non-vacuous. However, I would appreciate further clarification on how informative this asymptotic result is in practically relevant moderate-dimensional regimes. In particular, how quickly does the gap between the Gaussian mechanism and the true optimum shrink as $T$ increases? Could the authors provide additional finite-dimensional diagnostics—for example, the trend of the gap between the theoretical lower bound and the actual optimum across different values of $T$—or at least some empirical guidance on the dimensional range in which the asymptotic characterization already becomes predictive of observed behavior?
2. On the systematicity and robustness of the low-dimensional SGG improvements:
The paper introduces the SGG family, develops the associated privacy analysis, numerical calibration procedure, and parameter optimization method, and reports that in some low-dimensional settings it can achieve up to roughly 15% lower error than both the Gaussian and the $\ell_2$ mechanisms. I would appreciate more clarification on the extent to which these improvements are systematic, rather than concentrated in a relatively small subset of parameter regimes. For instance, could the authors provide a more complete sensitivity or robustness analysis showing how the optimal parameters $(\alpha, p, \beta)$ vary with $(\varepsilon,\delta,T)$, and how sensitive the reported gains are to perturbations of these parameters? Such an analysis would help clarify whether the low-dimensional contribution should be viewed as a broadly useful mechanism-design framework or primarily as an existence result demonstrating that nontrivial improvements are possible.

**Limitations:**

No.The authors do acknowledge some important boundaries of the current work. In particular, the conclusion makes clear that the present analysis is restricted to additive-noise mechanisms, and leaves data-dependent noise and non-additive mechanisms for future work. The Impact Statement also notes that differential privacy can be misused in practice, and that the choice of $(\varepsilon,\delta)$ should not be interpreted as a universally valid safety recommendation.
That said, I believe the discussion of limitations could still be strengthened in two respects.
First, the paper could more explicitly discuss the asymptotic nature of the high-dimensional result and clarify its relevance in finite-dimensional regimes, rather than relying primarily on numerical lower bounds for $\delta^\star$.
Second, in the discussion of broader impact, beyond the general statement that DP may be misused, the paper could more concretely emphasize that tighter accounting or lower noise does not automatically imply safe deployment; the actual privacy risk still depends on the underlying threat model, the chosen privacy parameters, and the application context. Although the paper already gestures in this direction, making this point more explicit would improve the discussion.

**Strengths And Weaknesses:**

1. Soundness
From a technical standpoint, this paper is fundamentally sound, with its core claims generally supported by the evidence presented. The main theorem in the dimensionality section is established through a comprehensive chain of proofs: the authors first reduce general additive noise to spherically symmetric noise via Haar symmetrization, then transform high-dimensional privacy comparisons into a one-dimensional problem concerning radial energy, and finally derive a Jensen-type lower bound using the support line property of Gaussian privacy functions, thereby achieving asymptotic optimality. Additionally, I believe the paper effectively demonstrates that its arguments are sufficiently supported. The scope of Theorem 3.1 is explicitly defined: it does not unconditionally claim Gaussian optimality in all finite-dimensional scenarios, but is restricted to the regime where $T\to\infty$, $\varepsilon$ and $s$ are fixed, and $\delta\le \delta^\star$. Numerical lower bounds are provided to demonstrate this regime is not empty, covering common choices where $\delta \le 10^{-3}$. For low-dimensional scenarios, the authors present an explicit one-dimensional integral form for the privacy expression of the SGG mechanism, establish monotonicity for the worst-case shift magnitude, and introduce an oracle with error control to ensure numerical calibration does not compromise DP guarantees.
However, several reservations remain regarding robustness. First, the high-dimensional optimality result relies on “strong privacy intervals” and asymptotic dimensionality assumptions, remaining fundamentally an asymptotic theorem. While the authors attempt to demonstrate practicality via a numerical lower bound for $\delta^\star$, the main text lacks further diagnostics on finite-dimensional convergence rates, error term scales, or when the theorem becomes “effective” at intermediate dimensions.
Second, verification of low-dimensional results currently relies primarily on numerical mechanism search and MSE comparisons. While the paper demonstrates “up to 15% improvement,” the evidence remains largely mechanism-level numerical comparisons like Figure 2, rather than application-oriented end-to-end DP learning/estimation experiments. Third, the authors themselves acknowledge in the conclusion that the current analysis remains confined to additive noise, reserving data-dependent noise and non-additive mechanisms for future work. While this boundary is reasonably justified, it also implies that the “optimality” claim is currently restricted to an important but specific class of mechanisms.
2. Presentation Style
The overall writing quality of the paper is quite high. The problem formulation is exceptionally clear: the introduction directly poses two core questions—whether Gaussian is optimal in high dimensions and whether a mechanism exists in low dimensions that simultaneously outperforms both Gaussian and $\ell_2$—with the subsequent organization of the paper largely structured around these two inquiries. For a theoretical paper, this narrative structure proves effective.
Additionally, the paper adeptly positions its relationship to prior work. The authors clearly distinguish between: optimality results under Gaussian DP, suboptimal work on pure DP/scalar queries, empirical studies on Generalized Gaussian families, and recent findings on $\ell_2$ mechanisms and composition. They thoroughly explain how their contributions differ from these existing approaches.
I also commend the authors' technical exposition: even when full proofs reside in the appendix, the main text's proof sketches convey key insights—such as why spherically symmetric transformations are necessary, why the problem reduces to a one-dimensional comparison of $g(R^2/T)$, and why the supporting-line property suffices. This ensures the paper delivers not just results, but also intuition.
3. Significance
The problem addressed in this paper is significant. Additive noise mechanisms represent one of the most fundamental and widely used primitives in DP, while Gaussian mechanisms are practically the default choice in implementation. The paper does not tackle a marginal issue, but rather seeks to determine when there exists a theoretical basis for optimality and when it is not the best approach—a question of both theoretical and practical value.
In terms of significance, I believe this work excels not merely by proving a theorem, but by simultaneously delivering two complementary conclusions: it provides justification for the widespread use of Gaussian mechanisms in high dimensions, while demonstrating in low dimensions that systematic improvement is still possible—not just outperforming Gaussian mechanisms, but simultaneously surpassing both Gaussian and recent $\ell_2$ mechanisms.
4. Originality
I find the originality of this paper to be clear and substantive. Its novelty lies primarily in the way it formulates the problem, the theoretical perspective it adopts, and the mechanism-design framework it develops. At a high level, the paper tackles two fundamental questions that, to my knowledge, have not previously been addressed in a unified manner: first, whether the Gaussian mechanism is asymptotically optimal among all additive-noise mechanisms in high dimensions under classical $(\varepsilon,\delta)$-DP at fixed MSE; and second, whether, in low dimensions, one can design mechanisms that outperform both the Gaussian mechanism and the recent $\ell_2$ mechanism.
The originality of the high-dimensional part lies not merely in showing that the Gaussian mechanism performs well, but in establishing its asymptotic optimality relative to a stronger and more natural benchmark: general additive-noise mechanisms under classical approximate DP, rather than a restricted mechanism class or an alternative formulation such as Gaussian DP. Technically, the paper proceeds by reducing the general case to the spherical setting via a symmetrization argument, and then converting the high-dimensional privacy comparison into a one-dimensional problem over radial energy. This yields a conceptually clean explanation for why the Gaussian mechanism emerges as optimal in the high-dimensional strong-privacy regime.
The low-dimensional contribution is also original, particularly in its unification and extension of prior mechanism families. The proposed SGG family is not simply an interpolation between the Gaussian and $\ell_2$ mechanisms, but a broader three-parameter spherically symmetric family that contains both as special cases. More importantly, the paper does not stop at proposing a more flexible family: it also develops the associated privacy analysis, numerical calibration machinery, parameter optimization procedure, and composition/accounting results, thereby turning the construction into a principled and practically usable mechanism-design framework. The availability of supplementary material, including code, further strengthens the completeness of the contribution.

---

> ### Author Rebuttal · Authors · 2026-03-31
>
> We thank the reviewer for the careful reading and thoughtful feedback. We are encouraged that the review generally views the paper as technically sound and recognizes the interest of its theoretical contributions. Below, we address the corresponding comments in more detail.
>
> ## Finite-dimensional relevance of the asymptotic result
> We thank the reviewer for this important question. The current draft does not provide a finite-dimensional convergence rate for how quickly the Gaussian’s asymptotic optimality kicks in.
> However, we believe such a quantitative refinement is plausible, and that the right place to extract it is Lemma 3.1.  We could replace the current $o(1)$ step in Lemma 3.1 by an explicit normal-approximation bound and convert that asymptotic approximation error to a concrete convergence rate through the test used in the proof. We believe a crude $O(T^{-1/2})$ convergence rate appears quite plausible with modest extra work, and an $O(T^{-1}) may well be achievable too. We leave this interesting question to future work.
>
> From the empirical evidence, we note that the current draft also gives directional evidence consistent with this picture in Fig 2: the advantage of the SGG mechanism over Gaussian mechanisms decreases sharply as T increases from 2 to 10 (and further experiments showed that by T=100, the MSE gap between SGG and Gaussian is already down to 1\%). We will add the discussion to make it clearer in the revision.
>
> ##  Systematicity and robustness of the low-dimensional SGG improvements
>
> We view the low-dimensional SGG contribution primarily as an existence result: in certain concrete low-dimensional settings, there are better mechanisms than  Gaussian/$\ell_2$ mechanism.
>
> At present, our evidence doesn’t support that SGG improves over Gaussian systematically, but rather concentrated in a small subset of parameter regimes. Additional experiments we deemed inconsequential for the paper showed that the SGG advantage shrinks rapidly with increasing dimension and only for less useful $\delta$ regimes; in particular, by around $T=100$, the improvement over Gaussian is just around 1/%. Furthermore, we leave it to future work to study the interesting question of how the $(\varepsilon, \delta)$ guarantees vary with parameters $(\alpha, p, \beta)$ and $T$.
>
> We will add this discussion and state more explicitly in the limitations section that the current SGG contribution is as a principled demonstration that nontrivial improvements are possible, rather than as evidence of a broadly uniform replacement for Gaussian across applications.
>
> ## Limitations
> We thank the reviewer for these helpful suggestions regarding limitations. We will strengthen both points the reviewer mentioned, and revise the draft accordingly.

---

> > ### Author Rebuttal · Reviewer_g1PK · 2026-04-03
> >
> > Thank you very much for addressing my comments carefully.  I will keep the original score as before.

---

### Official Review · Reviewer_aCxW · 2026-03-12

**Soundness:** 4
**Presentation:** 3
**Significance:** 2
**Originality:** 3
**Overall Recommendation:** 5
**Confidence:** 4

**Summary:**

This paper proposes a general class of spherically symmetric additive noise distributions and proves some optimality properties. In particular, Gaussian noise is asymptotically

I apologize for the terseness of this review. I had written something much more extensive but OpenReview failed to save it since there is no draft mode for reviews, even though this is 2026.

**Compliance With Llm Reviewing Policy:**

Affirmed.

**Final Justification:**

As I wrote initially, I think this paper is a shoo-in. It would be even stronger after adding the clarifications from the rebuttals to the various reviewers.

**Key Questions For Authors:**

The biggest challenge with this paper is that the implications of the results are not clear for theory or practice, despite the structural results being quite nice. On the theory side:

* The main result, Theorem 3.1, shows that for sufficiently large $T$ there is a "delta advantage" for Gaussian noise. Figure 4 is for much smaller $T$ and seems to suggest that the advantage decreases as $T$ increases, unless I am misinterpreting things. How do we make this make sense?

* Textbooks (c.f. Vershynin) often characterize results for isotropic distributions. SGG distributions are isotropic but are there subclasses of isotropic distributions which include SGG? There is such a zoo of distributions out there that it's possible that SGG exists under another name.

* The authors claim "realistic" $\delta \approx 10^{-3}$, which is perhaps true for experimental results on DP, where small $\delta$ kills utility. But theory says that we should choose $\delta \ll 1/n$ to be meaningful. How should we resolve this? Are the results more useful for theory or practice?

* The composition results are claimed in the main manuscript in an informal way but only appear in Appendix D. Reviewers are not required to review the supplementary materials and I think that a formal statement of the composition results should appear in the main manuscript.

* The composition results are FFT-based. Does that mean they are approximate?

If the results are most interesting for practice, it would be nice to see privacy-utility tradeoffs where optimizing the noise distribution from the generalized Gamma family leads to practical utility guarantees for the same privacy level. On the practice side:

* The experimental validation of the benefits of SGG mechanisms seems quite cursory in the main document and only a little more detailed in the supplement. Figure 2 suggests that the "dramatic" improvements decay with larger $T$.

* Since a key application for Gaussian noise is DP-SGD, where composition is a key tool, it would have been nice to see whether an SGG mechanism provides better utility from the learning-related utility perspective. That is, if we optimize for estimating the mean of clipped gradients for finite $T$, will we get a better learning method?

* [Comment not a question] The lower bounds on $\delta^*$, while useful for showing that the range of $\delta$ values is not vacuous, are perhaps less interesting and could be moved to the appendix if more space is needed

* [Vague and ignorable comment] Is there a way to frame this as $f$-DP?

**Limitations:**

Yes.

**Strengths And Weaknesses:**

**Strengths**

* The results are theoretically sound and provide new insight into the geometric structure of optimal noise mechanisms in the high-dimensional setting.
* The class of SGG distributions may be of broader interest.

**Weaknesses**

* The balance between practical and theoretical impacts are not clear.
* Composition results are claimed but not formally described in the main paper.

---

> ### Author Rebuttal · Authors · 2026-03-31
>
> We thank the reviewer for the careful reading and thoughtful feedback. We are encouraged that the review generally views the paper as technically sound and recognizes the interest of its theoretical contributions.
>
> We view the primary implications of this paper as theoretical/conceptual. Importantly, the main message of the paper (as also recognized by the reviewer) has two parts: i) asymptotically, in high dimensions, the Gaussian mechanism is optimal among additive mechanisms under MSE constraints, and ii) there exists room for improvement over the state-of-the-art mechanisms for certain concrete low dimensional settings; we provide a few concrete examples for low dimensions to show that there is a better mechanism than Gaussian/$\ell_2$ in Fig. 2. Note that while not particularly geared towards practice, the results in Fig. 2 can directly be used for DP-estimating the mean of clipped gradients (and thus DP-SGD), but for small T (and n) only. We will revise the paper to clearly explain the scope, while being explicit about what the present work does and does not show for interest in the practical side.
>
> Below, we address the corresponding comments in more detail.
>
> ## Theorem-3.1 and Figure-2
>
> Theorem-3.1 is asymptotic: for any fixed (\varepsilon,s) and sufficiently small \delta, it states that as T gets large enough, no additive mechanism with the same MSE can asymptotically beat Gaussian. It does not claim that Gaussian is optimal at any concrete T.
>
> Accordingly, the SGG's improvements in low-dimensions are not in tension with Theorem 3.1. Rather, they describe the finite-dimensional regime that the theorem leaves open: for small T, better additive mechanisms can exist; as T grows, the observed advantage shrinks (as shown in Fig 2), which is directionally consistent with Gaussian becoming asymptotically optimal.
>
> ## Privacy analysis of other isotropic / spherically symmetric families
>
> SGG is indeed not the most general isotropic/spherical family. What we believe is new is not the existence of a larger isotropic subclass, but the **unified DP treatment** of this particular subclass, which is broad enough to include the Gaussian mechanism, the $\ell_2$ mechanism, and R1SMG, and two-parameter generalized Gaussian family. To the best of our knowledge, we are not aware of any other DP treatment of distribution classes containing SGG.
>
> ## "realistic" delta
>
> We will revise the wording of "realistic". In the SGG study, the chosen value of $10^{-3}$ for delta is to show regimes where there remains nontrivial design space to improve utility over Gaussian/$\ell_2$ mechanisms. We do not try to claim that these are universally the “right” privacy parameters in real-world applications.
>
> ## SGG Composition
>
> We will revise the main body to include a formal statement of the composition result.
>
> SGG accounting guarantee is rigorous (not approximate): for any fixed $\varepsilon$, the procedure returns a valid numerical upper bound on $\delta$. The discretization and other numerical errors when applying FFT are explicitly absorbed into the final formal upper-bound margin.

---

> > ### Author Rebuttal · Reviewer_aCxW · 2026-03-31
> >
> > Looks good, thanks for the clarifications.

---

### Decision · Program_Chairs · 2026-04-30

**Decision:**

Accept (spotlight)

**Comment:**

This is a technically strong and conceptually valuable contribution. Reviewers agreed that the paper answers two important questions. First, it gives a new compelling justification for the use of Gaussian mechanism in high dimensions proving its asymptotic optimality among additive-noise mechanisms. Second, it defines a broader family of DP mechanisms (Spherical Generalized Gamma) that captures several known mechanisms, and that yields improvements in some low-dimensional regimes together with tight composition/accounting results. The claims are well-presented and the unifying perspective of the mechanism family seems insightful about the structure of optimal additive DP mechanisms.

The main concerns were about positioning the results rather than correctness. In particular, they centered around theoretical vs practical implications and how consistent the improvement of SGG in the low-dimensional regime is. The rebuttal addressed these points well.

I therefore recommend acceptance. The final version should include a discussion on the limits of the low-dimensional improvements (specifically that SGG does not consistently improve upon the Gaussian mechanism), as well as the note on finite-dimensional convergence.